# Bayesian Speech synthesizers Can Learn from Multiple Teachers

## Abstract

Codec-based text-to-speech (TTS) models have recently gained traction for their efficiency and strong performance in voice cloning. However, codec-based TTS faces limitations due to the challenges of pretraining robust speech codecs and the quality degradation introduced by quantization errors. Emerging evidence suggests that continuous-valued generative models can alleviate these issues and serve as a promising alternative. Yet, effectively modelling diverse speech patterns and developing reliable sampling strategies for continuous-valued autoregressive (AR) TTS remains underexplored. In this work, we propose BELLE, **B**ayesian **e**vidential **l**earning with **l**anguag**e** modelling for TTS, a novel continuous-valued AR framework that directly predicts mel-spectrograms from textual input. BELLE treats each mel-spectrogram frame as a Gaussian distribution sampled from a learned hyper-distribution, enabling principled uncertainty estimation, particularly in scenarios with parallel data (*i.e.*, one text-audio prompt paired with multiple speech samples). To obtain such data, diverse speech samples are synthesized using multiple pre-trained TTS models given the same text-audio prompts, which are distilled into BELLE via Bayesian evidential learning. Experimental results indicate that BELLE demonstrates highly competitive performance compared with the current best open-source TTS models, even though BELLE is trained on a large amount of synthetic data and uses only approximately one-tenth of their training data. Audio samples generated by BELLE are available at `https://belletts.github.io/Belle/`. The code, checkpoints, and synthetic data will be released after the paper is accepted.

## 1 Introduction

Recently, autoregressive (AR) next-token prediction models based on discrete audio codecs have gained significant attention in the audio generation and text-to-speech (TTS) communities, driven by the success of models such as AudioLM Borsos et al. (2023) and VALL-E (Wang et al., 2023), among others. However, continuous-valued prediction has emerged as a promising alternative due to its ability to eliminate quantization errors introduced by token-based codecs (Meng et al., 2024; Liu et al., 2024; Lin & He, 2025; Chen et al., 2024b; Zhu et al., 2024; Eskimez et al., 2024; Chen et al., 2024a; Wang et al., 2025a; Jia et al., 2025; Lee et al., 2025). Generative models typically rely on sampling mechanisms to introduce stochasticity and improve synthesis quality. In audio-codec-based AR TTS, this is often achieved using mature top-$p$ sampling techniques, similar to those employed in token-based language models (LMs). Diffusion-based models, by contrast, introduce randomness by injecting noise into the input, thereby embedding variability directly into the generation process. However, effective sampling strategies for continuous-valued AR models remain underexplored. Recent studies have begun to address this gap. In image generation, MAR proposed a hybrid approach where an AR model predicts conditioning signals, followed by diffusion-based sampling (Li et al., 2024a). In zero-shot TTS, MELLE introduced Gaussian sampling (Meng et al., 2024), while FELLE adopted flow-matching techniques (Lipman et al., 2022; Wang et al., 2025a) to enhance sampling efficiency and output diversity.

In this paper, we propose BELLE: **B**ayesian **e**vidential **l**earning with **l**anguag**e** modelling for TTS Synthesis—a continuous-valued AR model that directly predicts mel-spectrograms from text inputs based on the evidential deep learning (EDL) framework (Amini et al., 2020). This represents the first application of Bayesian deep learning to TTS, enabling controllable and theoretically grounded

sampling at test time. Specifically, BELLE models each predicted mel-spectrogram frame as a sample from a Gaussian distribution, whose parameters themselves are drawn from hyper-distributions. Metaphorically, the model first predicts the parameters of this hyper-distribution, then samples a specific Gaussian distribution from it, and finally generates mel-spectrogram frames from the sampled Gaussian. However, learning accurate Gaussian posterior estimates from limited data poses a challenge, as typical TTS datasets contain only a single audio recording per text prompt. To overcome this, the training corpus is augmented with synthetic audio samples generated by several publicly available pre-trained zero-shot TTS models (Du et al., 2024; Deng et al., 2025; Wang et al., 2025c; 2024; Casanova et al., 2024).

Our main contributions are as follows:

- We present BELLE, the first AR TTS model using evidential deep learning to train and sample from continuous-valued Mel-Spectrogram space. The results show that Bayesian evidential sampling outperforms conventional Gaussian sampling.
- We propose the first multi-teacher knowledge distillation approach for TTS model training, integrating it with evidential deep learning.
- BELLE achieves competitive performance compared with leading open-source TTS systems trained on approximately 50k hours of speech data, while BELLE uses only around 5k hours of data that largely consist of synthetic samples. Additionally, its streaming variant, BELLE-stream, attains an effective balance between audio generation quality and latency.

## 2 RELATED WORK

**Continuous-valued Zero-Shot TTS and Streaming TTS.** Continuous-valued TTS models can be broadly categorized into AR and non-autoregressive (NAR) approaches. AR methods include MELLE (Meng et al., 2024), which generates mel spectrogram frames using decoder-only Transformer-based AR modelling combined with Gaussian sampling. FELLE (Wang et al., 2025a), ARDiT (Liu et al., 2024) and DiTAR (Jia et al., 2025) first generate continuous-valued hidden states autoregressively, then employ flow-matching or diffusion-based modules, such as DiT (Peebles & Xie, 2023), to sample and produce mel spectrograms or a continuous latent space learned by a VAE (Pinheiro Cinelli et al., 2021). Moreover, VAE-based AR methods, including GMM-LM (Lin & He, 2025) and KALLE (Zhu et al., 2024), initially leverage a VAE to estimate a latent distribution (mean and variance), and subsequently autoregressively predict latent representations sampled from this distribution. NAR models such as E2 (Eskimez et al., 2024) and F5 (Chen et al., 2024a), on the other hand, predict mel-spectrograms directly via iterative mask-based refinement. Recently, several streaming TTS systems have emerged (Du et al., 2024; Yang et al., 2024; Sun et al., 2025; Sheng et al., 2025; Wang et al., 2025b). Most of these approaches model discrete features using a language model, followed by a flow-matching module or other similar components to generate speech features. Such architectures are relatively complex and may introduce potentially higher latency.

**Bayesian Methods in Sequence Modelling and TTS.** Evidential Deep Learning (EDL) (Amini et al., 2020) is commonly used in recurrent neural networks (RNNs) and Transformers to provide per-time-step uncertainty (Cui et al., 2024; Khot et al., 2024; Marvi et al., 2025), benefiting tasks like time-series forecasting (Khot et al., 2024; Marvi et al., 2025). Other Bayesian approaches, such as Bayesian neural networks (Kononenko, 1989) in Long Short-Term Memory (LSTMs) and Transformers (Muñoz et al., 2024), and frameworks like SYMHnet (Abduallah et al., 2024), also address sequential uncertainty. Previous Bayesian TTS work focused on HMM-based models (Hashimoto et al., 2009b;a; Lu & King, 2012); to our knowledge, we are the first to apply Bayesian methods to neural network-based TTS.

**Teacher-Student Training in TTS.** Knowledge distillation (KD) transfers knowledge from larger teacher models to smaller, faster student models (Nguyen et al., 2025; Yang et al., 2022; Li et al., 2024b). Originally, KD is used to reduce exposure bias in AR models like Tacotron (Wang et al., 2017; Liu et al., 2020) and to guide NAR models such as FastSpeech (Ren et al., 2019; 2020). It has been further applied to improve pronunciation (Nguyen et al., 2025), style transfer (Yang et al., 2022), perceptual quality in diffusion-based models (Li et al., 2024b), and to distill semantic knowledge

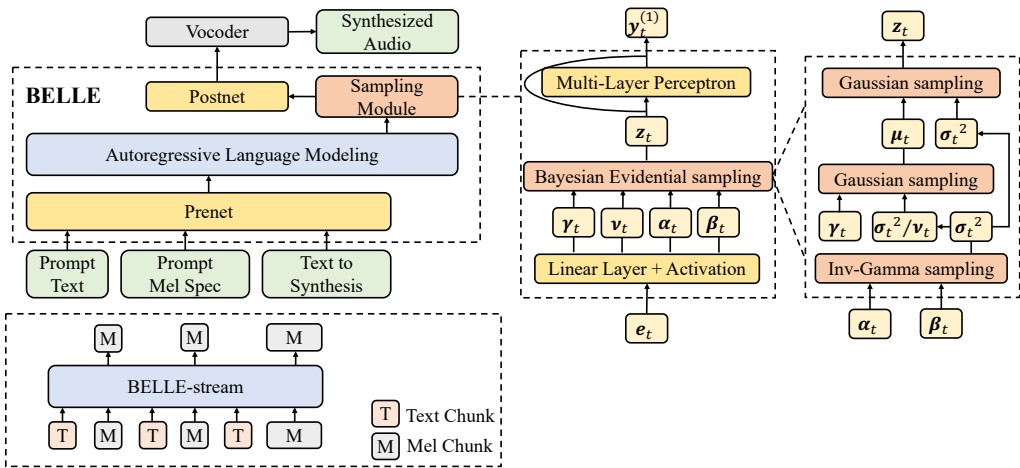

Figure 1: The structure of BELLE and the detailed sampling module. The output is assumed following a Normal-Inverse-Gamma (NIG) distribution and Sampling Module predicts four distribution parameters. Sequentially, variance and mean are obtained via Inverse-Gamma sampling and Gaussian sampling, respectively, followed by a final Gaussian sampling step to generate the output $\boldsymbol{y}_t^{(1)}$. BELLE-stream adopts the same architecture as BELLE. The text and Mel-spectrogram are split into chunks and interleaved to form the input sequence. Typically, the final chunk generates all remaining audio, making it longer than the preceding ones.

from HuBERT (Gállego et al., 2025). To our knowledge, multi-teacher KD has not yet been explored in TTS.

## 3 MEL-BASED AUTOREGRESSIVE TTS

### 3.1 PROBLEM FORMULATION

AR TTS models based on mel-spectrogram synthesis typically formulate the generation process as a sequential next-frame prediction task. Formally, given an input text sequence $\boldsymbol{x} = [x_1, x_2, ..., x_N]$, it's aimed at generating an acoustic mel-spectrogram sequence $\boldsymbol{y} = [\boldsymbol{y}_1, \boldsymbol{y}_2, ..., \boldsymbol{y}_T]$, where each frame $\boldsymbol{y}_t \in \mathbb{R}^D$ denotes the spectral representation at time step $t$, and $D$ represents the number of mel-frequency bands. In AR modeling, each mel-spectrogram frame $\boldsymbol{y}_t$ is generated conditionally depending on the textual content $\boldsymbol{x}$ and previous frames $\boldsymbol{y}_{<t} = [\boldsymbol{y}_1, ..., \boldsymbol{y}_{t-1}]$. Thus, the generative process can be described by the following conditional probability decomposition:

$$p(\boldsymbol{y}|\boldsymbol{x};\theta) = \prod_{t=1}^{T} p(\boldsymbol{y}_t|\boldsymbol{y}_{<t}, \boldsymbol{x};\theta) \tag{1}$$

where the conditional probability is typically modelled by a deep learning model with parameters $\theta$.

The **streaming generation task** is defined that both the input text and target mel-spectrogram are segmented into $M$ consecutive chunks $\boldsymbol{X} = [\boldsymbol{x}^{(1)}, \ldots, \boldsymbol{x}^{(M)}]$ and $\boldsymbol{Y} = [\boldsymbol{y}^{(1)}, \ldots, \boldsymbol{y}^{(M)}]$. At step $m$, the current audio chunk $\boldsymbol{y}^{(m)}$ is predicted conditioned on all previously generated audio chunks $\boldsymbol{y}^{(<m)}$ and all available text chunks $\boldsymbol{x}^{(\leq m)}$:

$$p(\boldsymbol{Y} \mid \boldsymbol{X};\theta) = \prod_{m=1}^{M} p(\boldsymbol{y}^{(m)} \mid \boldsymbol{y}^{(<m)}, \boldsymbol{x}^{(\leq m)};\theta). \tag{2}$$

### 3.2 GENERAL ARCHITECTURE OF MEL-BASED AUTOREGRESSIVE TTS MODEL

General mel-based AR TTS model consists of several interconnected modules: a **Prenet**, an **AR LM**, a **Sampling Module**, a **Postnet**, a **Stop-Prediction Module**, and a **Vocoder**, which are shown in Fig. 1. These modules will be introduced in the following sections.

### 3.2.1 PRENET AND CONTINUOUS-VALUED AR LM

The continuous-valued AR generation model closely resembles standard LM architectures, specifically decoder-only Transformer models that are frequently employed in contemporary large language models (LLMs), which is shown in the left part of Fig. 1. To delineate the input text clearly, a special $\langle\text{BOS}\rangle$ token is added at the beginning of textual sequences, and an $\langle\text{EOS}\rangle$ token is appended at the end. The Prenet then maps discrete textual tokens $\boldsymbol{x}$ into continuous embeddings as well as maps mel-spectrogram frames $\boldsymbol{y}$ to the hidden dimension of the AR LM. Next, the AR LM accepts a concatenation of textual embeddings and mel-spectrogram embeddings as inputs, producing corresponding hidden representations $\boldsymbol{e}_t$.

### 3.2.2 SAMPLING MODULE

The Sampling Module introduces critical stochasticity into the generative process. Specifically, the hidden states $\boldsymbol{e}_t$ from the AR LM are first projected down to the mel-spectrogram dimension. The projected representation is assumed to follow a specified parametric distribution (*e.g.*, a Gaussian distribution). Subsequently, the Sampling Module samples from this distribution, obtaining an intermediate sampled representation denoted as $\boldsymbol{z}_t$. Afterwards, a multilayer perception (MLP)-based denoising network refines and further processes the sampled representation, and the resulting denoised output is denoted as $\boldsymbol{y}_t^{(1)}$ as shown in the right part of Fig. 1. The detailed description of our proposed evidential Bayesian sampling method is provided in Sec. 4.

### 3.2.3 STOP-PREDICTION MODULE AND POSTNET

Discrete-token AR LM typically incorporates an explicit $\langle\text{EOS}\rangle$ token prediction to naturally terminate sequence generation. For continuous-valued AR generation, a dedicated Stop-Prediction Module is required. Following TransformerTTS or SpeechT5 (Li et al., 2019; Ao et al., 2022), this module predicts a scalar value through a single linear layer followed by a sigmoid activation function, resulting in a score between 0 and 1. Generation stops when this normalized score exceeds a predefined threshold.

The Postnet consists of multiple convolutional layers to refine generated acoustic features like methods in (Shen et al., 2018; Li et al., 2019). Given the output of the Sampling Module $\boldsymbol{y}_t^{(1)}$, the Postnet predicts a residual term, which is subsequently added back to $\boldsymbol{y}_t^{(1)}$ and produces the refined output $\boldsymbol{y}_t^{(2)}$. During training, the model employs the teacher-forcing paradigm; at inference, all intermediate $\boldsymbol{y}_t^{(1)}$ frames are first collected and then passed through the Postnet to obtain the final output $\boldsymbol{y}_t^{(2)}$.

### 3.2.4 STREAMING CHUNK-BASED GENERATION

As shown in Fig. 1, a streaming generation strategy by segmenting both the input text and target acoustic sequence into fixed-size chunks is adopted. Let the predefined text chunk size be denoted as $S_{\text{text}}$ and the audio chunk size as $S_{\text{audio}}$. The text sequence is first partitioned into chunks $\boldsymbol{x}^{(1)}, \ldots, \boldsymbol{x}^{(M)}$, where the last chunk may contain fewer than $S_{\text{text}}$ tokens. Similarly, the mel-spectrogram is partitioned into $\boldsymbol{y}^{(1)}, \ldots, \boldsymbol{y}^{(M)}$, each containing up to $S_{\text{audio}}$ frames, with the last chunk possibly exceeding $S_{\text{audio}}$ frames.

The generation process interleaves text and audio chunks in the input to the AR LM, maintaining a causal mask over the sequence. By setting a relatively smaller $S_{\text{audio}}$, each audio chunk is generated with sufficient preceding textual context to ensure coherent synthesis. At inference time, chunk-by-chunk audio generation is performed until the last audio chunk, where the Stop-Prediction Module is invoked to decide the termination point of synthesis.

## 4 BAYESIAN EVIDENTIAL LEARNING

### 4.1 PRELIMINARY: EDL IN MEL-SPECTROGRAM PREDICTION

EDL is a Bayesian approach for uncertainty quantification in regression tasks by explicitly modelling the posterior distribution of predictions. In our TTS setting, the observed data $\boldsymbol{y}_t$ denotes the mel-spectrogram frame at time step $t$, where $\boldsymbol{y}_t \in \mathbb{R}^D$.

Each frame is assumed to follow a Gaussian distribution with unknown mean $\boldsymbol{\mu}_t$ and variance $\boldsymbol{\sigma}_t^2$, with an Normal-Inverse-Gamma (NIG) conjugate prior:

$$\boldsymbol{y}_t \sim \mathcal{N}(\boldsymbol{\mu}_t, \boldsymbol{\sigma}_t^2), \quad \boldsymbol{\mu}_t \sim \mathcal{N}\left(\boldsymbol{\gamma}_t, \frac{\boldsymbol{\sigma}_t^2}{\boldsymbol{\nu}_t}\right), \quad \boldsymbol{\sigma}_t^2 \sim \Gamma^{-1}(\boldsymbol{\alpha}_t, \boldsymbol{\beta}_t), \tag{3}$$

where all hyperparameters $\boldsymbol{\gamma}_t, \boldsymbol{\nu}_t, \boldsymbol{\alpha}_t, \boldsymbol{\beta}_t \in \mathbb{R}^D$. The constraints are applied element-wise: $\nu_{t,d} > 0, \alpha_{t,d} > 1, \beta_{t,d} > 0, \forall d \in \{1, \dots, D\}$.

The posterior predictive distribution is then a multivariate Student-$t$ with diagonal scale:

$$p(\boldsymbol{y}_t|\boldsymbol{\gamma}_t, \boldsymbol{\nu}_t, \boldsymbol{\alpha}_t, \boldsymbol{\beta}_t) = \text{St}\left(\boldsymbol{y}_t; \boldsymbol{\gamma}_t, \frac{\boldsymbol{\beta}_t \odot (1 + \boldsymbol{\nu}_t)}{\boldsymbol{\nu}_t \odot \boldsymbol{\alpha}_t}, 2\boldsymbol{\alpha}_t\right), \tag{4}$$

where $\odot$ denotes element-wise multiplication.

In practice, the hyperparameters $(\boldsymbol{\gamma}_t, \boldsymbol{\nu}_t, \boldsymbol{\alpha}_t, \boldsymbol{\beta}_t)$ are predicted by the sampling module and optimized via the evidential loss:

$$\mathcal{L}_{\text{edl}}(\boldsymbol{y}_t) = \mathcal{L}_{\text{NLL}}(\boldsymbol{y}_t) + \lambda \, \mathcal{L}_{\text{R}}(\boldsymbol{y}_t), \tag{5}$$

where the NLL enforces distributional fit and the regularization term penalizes incorrect evidence. Details about EDL are in Appendix D.

## 4.2 FROM GAUSSIAN SAMPLING TO BAYESIAN EVIDENTIAL SAMPLING

MELLE (Meng et al., 2024) directly predicts the mean and variance of a Gaussian distribution, then samples embeddings accordingly; implementation details are in Appendix E.1. We propose to replace the Gaussian sampling mechanism with a Bayesian evidential approach. Specifically, NIG distribution parameters are predicted from the AR hidden representation $\boldsymbol{e}_t$ via a linear layer followed by suitable activation functions to constrain each parameter within appropriate numerical ranges:

$$[\boldsymbol{\gamma}_t, \boldsymbol{\nu}_t, \boldsymbol{\alpha}_t, \boldsymbol{\beta}_t] = \text{Activation}(\boldsymbol{W}_{\text{NIG}}\boldsymbol{e}_t + \boldsymbol{b}_{\text{NIG}}), \tag{6}$$

where $\boldsymbol{\gamma}_t, \boldsymbol{\nu}_t, \boldsymbol{\alpha}_t, \boldsymbol{\beta}_t \in \mathbb{R}^D$, and Activation$(\cdot)$ is the selected activation function. Here, suitable activation functions (e.g., softplus for $\boldsymbol{\nu}_t, \boldsymbol{\beta}_t$, and softplus-shifted for $\boldsymbol{\alpha}_t$) are applied to ensure that they satisfy the numerical constraints of the NIG distribution (See Sec. 4.1).

During sampling, variance $\boldsymbol{\sigma}_t^2$ is first drawn dimension-wise from an Inverse-Gamma distribution parameterized by $\boldsymbol{\alpha}_t$ and $\boldsymbol{\beta}_t$. Conditioned on the sampled variance, the mean $\boldsymbol{\mu}_t$ is subsequently drawn from a Gaussian distribution with mean $\boldsymbol{\gamma}_t$ and variance $\boldsymbol{\sigma}_t^2/\boldsymbol{\nu}_t$. Finally, conditioned on the sampled parameters $\boldsymbol{\mu}_t$ and $\boldsymbol{\sigma}_t^2$, $\boldsymbol{z}_t$ is sampled from another Gaussian distribution with mean $\boldsymbol{\mu}_t$ and variance $\boldsymbol{\sigma}_t^2$. The overall hierarchical sampling procedure is summarized as:

$$\boldsymbol{\sigma}_t^2 \sim \Gamma^{-1}(\boldsymbol{\alpha}_t, \boldsymbol{\beta}_t), \quad \boldsymbol{\mu}_t \sim \mathcal{N}\left(\boldsymbol{\gamma}_t, \frac{\boldsymbol{\sigma}_t^2}{\boldsymbol{\nu}_t}\right), \quad \boldsymbol{z}_t \sim \mathcal{N}\left(\boldsymbol{\mu}_t, \boldsymbol{\sigma}_t^2\right). \tag{7}$$

## 4.3 TRAINING OBJECTIVES

BELLE is trained end-to-end using a composite loss:

$$\mathcal{L} = \mathcal{L}_{\text{reg}} + \lambda_{\text{samp}}\mathcal{L}_{\text{samp}} + \lambda_{\text{flux}}\mathcal{L}_{\text{flux}} + \mathcal{L}_{\text{stop}}, \tag{8}$$

where $\lambda_{\text{samp}}$ and $\lambda_{\text{flux}}$ balance the contribution of sampling and flux losses.

The regression loss $\mathcal{L}_{\text{reg}}$ ensures predicted mel-spectrograms closely match the ground truth $\boldsymbol{y}^{\text{gt}}$. It includes L1 and L2 terms for both coarse predictions $\boldsymbol{y}^{(1)}$ and refined predictions $\boldsymbol{y}^{(2)}$:

$$\mathcal{L}_{\text{reg}} = \sum_{j=1}^2 \left(||\boldsymbol{y}^{\text{gt}} - \boldsymbol{y}^{(j)}||_1 + ||\boldsymbol{y}^{\text{gt}} - \boldsymbol{y}^{(j)}||_2^2\right). \tag{9}$$

The sampling loss $\mathcal{L}_{\text{samp}}$ is used as the EDL loss:

$$\mathcal{L}_{\text{samp}}(\boldsymbol{y}^{\text{gt}}) = \mathcal{L}_{\text{edl}}(\boldsymbol{y}^{\text{gt}}) = \mathcal{L}_{\text{NLL}}(\boldsymbol{y}^{\text{gt}}) + \lambda\mathcal{L}_{\text{R}}(\boldsymbol{y}^{\text{gt}}), \tag{10}$$

whereas MELLE employs a KL loss for sampling.

The spectrogram flux loss $\mathcal{L}_{\text{flux}}$ encourages temporal dynamics, promoting variability between the current predicted distribution location $\boldsymbol{\gamma}_t$ and previous-frame ground truth $\boldsymbol{y}_{t-1}^{\text{gt}}$:

$$\mathcal{L}_{\text{flux}} = -\sum_{t=1}^{T-1} ||\boldsymbol{\gamma}_t - \boldsymbol{y}_{t-1}^{\text{gt}}||_1. \tag{11}$$

In contrast, MELLE uses its Gaussian mean $\boldsymbol{\mu}_t$.

Lastly, the stop prediction loss $\mathcal{L}_{\text{stop}}$ is a binary cross-entropy loss applied to stop logits from the hidden state $\boldsymbol{e}_t$. Due to class imbalance, the stop frame receives a higher weight of 500.

### 4.4 Multi Teacher Learning

A couple of open-sourced TTS models are used to generate multiple audio samples given the texts provided by the dataset. Given a textual input $\boldsymbol{x}$, audio samples are synthesized using a set of $N$ external TTS teacher models, resulting in $N$ synthesized mel-spectrograms. Together with the original human-recorded mel-spectrogram from the dataset, a total of $N+1$ corresponding mel-spectrograms are gotten. Denote these ground-truth mel-spectrograms as $\boldsymbol{y}_i^{\text{gt}}$, $i = 1, 2, \ldots, N+1$. Each mel-spectrogram is treated as a ground-truth example and calculated an individual loss corresponding to each one. To balance the contributions from multiple teachers and the original dataset recording, the predefined weights $w_i$ are assigned to each ground-truth mel-spectrogram.

Let $\mathcal{L}(\boldsymbol{y}_i^{\text{gt}})$ denote the complete loss computation defined in Eqn. (8) using $\boldsymbol{y}_i^{\text{gt}}$ as the ground-truth mel-spectrogram. The final overall training loss is computed as a weighted sum:

$$\mathcal{L}_{\text{multi-teacher}} = \sum_{i=1}^{N+1} w_i \mathcal{L}(\boldsymbol{y}_i^{\text{gt}}), \quad \text{with} \quad \sum_{i=1}^{N+1} w_i = 1, \quad w_i \geq 0. \tag{12}$$

## 5 Experimental Setup

### 5.1 Training Data and Details

The training dataset is derived from the Librispeech (Panayotov et al., 2015) training set and contains audio samples whose duration is from 0.5 second to 14 seconds, creating a final training dataset containing approximately 706 hours of speech. To provide richer acoustic diversity necessary for robust Bayesian distribution estimation, our training data are augmented by synthesizing multiple audio samples for each textual input using six publicly available pretrained TTS models, namely CosyVoice2 (Du et al., 2024), IndexTTS (Deng et al., 2025), SparkTTS (Wang et al., 2025c), F5TTS (Chen et al., 2024a), MaskGCT (Wang et al., 2024), and XTTS-v2 (Casanova et al., 2024). Including the TTS-synthesized data, the total dataset amounts to 4,817 hours of speech. Details of the data processing methodology can be found in Appendix C.1.

BELLE, BELLE-stream and MELLE are trained on the training dataset, following the training strategy of combining various data sources within each batch (see Sec. 4.4). For BELLE-stream, we set $S_{\text{text}} = 20$ and $S_{\text{audio}} = 50$, corresponding to an audio chunk duration of approximately 0.8 seconds. It's observed that the synthesis quality deteriorates with speech prompt. To address this, we first investigate the potential of BELLE for streaming synthesis in a single-speaker scenario, finetuning BELLE-stream using recordings from a single speaker of Librispeech to fix its voice timbre. Details about model configuration and training could be found in Appendix C.2 and C.3 respectively.

### 5.2 Evaluation Settings

The zero-shot TTS capabilities of our model is evaluated using the LibriSpeech test-clean subset by an open-source evaluation protocol (Lee, 2024). Specifically, two inference conditions are considered: **Continuation**, in which the first 3 seconds of an utterance and its corresponding transcription serve as the prompt, and the model synthesizes the continuation of speech thereafter; and **Cross-sentence**, where a reference utterance and its corresponding transcription from a given speaker is used as a prompt, and then the model generates speech for a different sentence.

For objective evaluation, word error rate (WER) is reported to measure intelligibility and robustness. WER-C and WER-H are evaluated using Conformer(Gulati et al., 2020) and HuBERT(Hsu

Table 1: Comparison of MOS and SMOS for different systems. BELLE-stream is fixed to a single speaker timbre and does not use an audio prompt, hence SMOS is not reported.

| System | Ground Truth | MaskGCT | F5-TTS | MELLE | BELLE | BELLE-stream |
|--------|-------------|---------|--------|-------|-------|--------------|
| MOS | 4.20 | 4.12 | **4.25** | 4.02 | 4.21 | 4.06 |
| SMOS | 3.81 | 3.89 | 3.95 | 3.80 | **4.13** | |

Table 2: Comparison of WER (%) and speaker similarity metrics for BELLE and baselines. Entries marked with † are trained using the dataset described in Sec. 5.1, marked with * correspond to the streaming TTS setting. BELLE-stream only takes text as input and does not use audio prompt, different from Cross-Sentence setting, so SIM metrics are omitted.

| System | Continuation | | | | Cross-Sentence | | | |
|--------|-------|-------|-------|-------|-------|-------|-------|-------|
| | WER-C | WER-H | SIM-r | SIM-o | WER-C | WER-H | SIM-r | SIM-o |
| Ground Truth | 1.78 | 2.15 | - | 0.668 | 1.78 | 2.15 | - | 0.672 |
| VALL-E | - | 3.8 | 0.508 | - | - | 5.9 | 0.580 | - |
| MaskGCT | - | - | - | - | - | **2.63** | - | **0.687** |
| CLAM-TTS(Kim et al., 2024) | - | 2.36 | 0.513 | 0.477 | - | 5.11 | 0.538 | 0.495 |
| MELLE † | 2.04 | 2.59 | 0.526 | 0.488 | 3.30 | 3.83 | 0.652 | 0.606 |
| SMLLE(Sun et al., 2025) * | - | - | - | - | 5.14 | 6.37 | 0.516 | 0.489 |
| IST-LM(Yang et al., 2024) * | - | 3.60 | - | - | - | 4.53 | - | 0.653 |
| BELLE † | **1.63** | **2.13** | **0.549** | **0.519** | **2.45** | 2.99 | **0.679** | 0.641 |
| BELLE-stream* | - | - | - | - | 2.70 | 3.97 | - | - |

et al., 2021) based ASR models respectively. Speaker similarity is measured via cosine similarity of extracted speaker embeddings, with SIM-o referencing the original speech prompt and SIM-r referencing the vocoder-reconstructed prompt.

For subjective evaluation, MOS and SMOS scores are obtained via a crowd-sourcing platform. MOS for evaluating overall speech quality, and SMOS for measuring speaker similarity between the generated audio and the prompt. MOS and SMOS is evaluated following the detailed procedure described in Appendix F.

For diversity evaluation, building on the layer-wise analysis of WavLM(Chiu et al., 2025), it's found that the middle-layer features contain rich paralinguistic information, which can be leveraged to assess the acoustic characteristics of speech. Frame-level hidden states from layer-13 are extracted for each sample, mean-pooled and L2-normalized, followed by computation of within-group pairwise cosine similarity and L1/L2 distances.

All evaluation details could be found in Appendix C.5.

# 6 RESULTS AND DISCUSSIONS

## 6.1 MAIN RESULTS

From Table 1, BELLE, F5-TTS, and Ground Truth achieve MOS scores of around 4.2, indicating that the audio produced by both F5-TTS and BELLE is close to that of natural human speech. F5-TTS attains a slightly higher MOS, which can be attributed to its lower background noise — human raters tend to prefer cleaner audio. In contrast, the real speech in Ground Truth inherently contains a certain level of noise, and BELLE replicates this natural background characteristic, leading to MOS scores that are very close to the Ground Truth. Regarding SMOS, it is worth noting that in the LibriSpeech dataset, different utterances from the same speaker sometimes exhibit perceptible timbre variation, which results in a relatively lower SMOS score for Ground Truth. BELLE achieves the highest speaker similarity among all tested systems. Notably, while both F5-TTS and MaskGCT were trained on datasets comprising 50,000 hours of English speech, BELLE was trained on less than 5,000 hours of data, yet it attained competitive, and in some instances superior performance. These

Table 3: Comparison of diversity metrics (cosine distance, L1, and L2) under the Cross-sentence setting. Each of 1,234 LibriSpeech test-clean prompts is generated three times; metrics are computed as within-prompt pairwise distances using WavLM-Large (layer 13) embeddings. BELLE ($\beta * 2$) doubles the sampling $\beta$ relative to the default to encourage diversity.

| System | Cosine distance | L1 | L2 |
|---|---|---|---|
| BELLE ($\beta * 2$) | **0.0053** | **3.04** | **0.095** |
| BELLE | 0.0037 | 2.62 | 0.080 |
| MELLE | 0.0033 | 2.55 | 0.078 |
| F5-TTS | 0.0005 | 0.40 | 0.012 |

results underscore the efficacy of the proposed Bayesian sampling strategy and the multi-teacher learning framework.

For the objective metrics in Table 2, BELLE attains the best WER and SIM scores under the Continuation setting. Under the Cross-sentence setting, MaskGCT achieves slightly higher scores; however, in our reproduction, MaskGCT's WER-C is only 3.86% under our evaluation protocol. This suggests that BELLE remains highly competitive in the Cross-sentence scenario. Additionally, BELLE outperforms MELLE across both subjective and objective metrics, including MOS, SMOS, WER, and SIM. These results demonstrate that Bayesian sampling delivers superior performance compared to Gaussian sampling.

Since BELLE-stream is finetuned to a fixed speaker timbre, it does not accept an audio prompt; therefore, it is excluded from SMOS scoring in the subjective evaluation and SIM computation in the objective evaluation, and its results are reported only under the Cross-sentence setting. BELLE-stream only takes the target text to be synthesized as input. From Table 1, streaming generation does not cause significant degradation in speech naturalness, achieving higher MOS scores than non-streaming MELLE. From Table 2, BELLE-stream achieves the lowest WER among streaming TTS systems, with no appreciable increase in WER compared to BELLE. In our tests, BELLE-stream achieved a real-time factor (RTF)—defined as the ratio of audio generation time to the length of the generated audio—of 0.55 when producing a 10-second utterance. Given that BELLE-stream operates with 0.8-second chunks, this implies a first-packet latency (FPL) of only 440ms, demonstrating its effective balance between performance and latency.

## 6.2 DIVERSITY ANALYSIS

The diversity evaluation is conducted using the evaluation method showed in Sec. 5.2. In the NIG prior, the parameter $\beta_t$ denotes the scale of the inverse-gamma distribution over the variance $\sigma_t^2$. Increasing $\beta_t$ raises the expected variance, thereby promoting greater diversity in the sampled outcomes (See Eqn. (7)). From Table 3, it's observed that BELLE ($\beta * 2$) exhibits the largest distances across all three metrics, confirming that increasing the sampling parameter $\beta$ effectively enhances generation diversity. Default BELLE achieves moderate diversity, slightly higher than MELLE, suggesting that Bayesian sampling yields superior diversity compared to Gaussian sampling. In contrast, F5-TTS produces consistently small distances, indicating that its outputs are more deterministic and less diverse across repeated sampling. These findings highlight that, beyond intelligibility and speaker similarity, BELLE offers a controllable trade-off between stability and diversity at inference time, with the $\beta$ parameter serving as a practical knob to adjust sample variability, analogous to the role of the temperature parameter in the sampling process of token-based language models.

## 6.3 THE EFFECT OF MULTI TEACHERS

Due to limited computational resources, the analysis experiments are conducted on a smaller-scale dataset. The *1-teacher* setting trains solely on the original Librispeech data without synthesized augmentation. The *3-teacher* setting incorporates additional synthesized speech from two TTS models, while the *7-teacher* setting augments the training data with speech from all six TTS models described earlier. The *data-aug* configuration serves as a baseline using conventional data augmentation, randomly sampling from all available data sources during training. Details about the small dataset and model training configuration could be found in Appendix C.4.

Table 4: Comparison of WER (%) and speaker similarity metrics for BELLE under different conditions trained on a smaller-scale dataset. *1-teacher* indicates training with only the original Librispeech dataset; *3-teacher* additionally includes XTTS-v2 and MaskGCT synthesized data; *7-teacher* further includes synthesized data from six TTS models. *data-aug* randomly samples from combined all data sources of 7 teachers.

| System | Continuation | | | | Cross-Sentence | | | |
|---|---|---|---|---|---|---|---|---|
| | WER-C | WER-H | SIM-r | SIM-o | WER-C | WER-H | SIM-r | SIM-o |
| BELLE *1-teacher* | 3.10 | 3.92 | 0.344 | 0.303 | 6.11 | 7.34 | 0.374 | 0.330 |
| BELLE *3-teacher* | 2.04 | 2.66 | 0.398 | 0.362 | **4.12** | 4.84 | 0.433 | 0.395 |
| BELLE *7-teacher* | **1.96** | **2.57** | **0.444** | **0.409** | 4.17 | **4.73** | **0.485** | **0.449** |
| BELLE *data-aug* | 2.10 | 2.64 | 0.439 | 0.404 | 4.83 | 5.39 | 0.480 | 0.443 |

Table 5: Ablation study examining the effects of the sampling module and flux loss on BELLE *3-teacher*. ✓ indicates enabled, ✗ indicates disabled.

| Sampling | Flux Loss | Continuation | | | | Cross-Sentence | | | |
|---|---|---|---|---|---|---|---|---|---|
| | | WER-C | WER-H | SIM-r | SIM-o | WER-C | WER-H | SIM-r | SIM-o |
| ✗ | ✓ | 4.30 | 5.00 | 0.382 | 0.344 | 12.04 | 13.07 | 0.392 | 0.351 |
| ✓ | ✗ | 2.50 | 3.25 | 0.388 | 0.347 | 5.61 | 6.37 | 0.417 | 0.372 |
| ✓ | ✓ | **2.04** | **2.66** | **0.398** | **0.362** | **4.12** | **4.84** | **0.433** | **0.395** |

As shown in Table 4, the performance of BELLE consistently improves as the number of teachers increases. Furthermore, the possibility that the performance gain is solely due to the increased amount of training data is excluded. In the *data-aug* setting, the total training data is larger than that of the *3-teacher* configuration; however, its performance is inferior to *3-teacher* on multiple metrics. This degradation may be attributed to the conventional data augmentation strategy, in which the same text can correspond to acoustically different speeches, potentially causing confusion for the model. In contrast, under our proposed multi-teacher training scheme, increasing the number of teachers leads to consistent performance gains, thereby demonstrating the effectiveness of the multi teacher training strategy.

### 6.4 THE EFFECTS OF THE SAMPLING MODULE AND FLUX LOSS

An ablation study is conducted to investigate how the sampling module and flux loss affect the performance of the BELLE model. Due to limited computational resources, the analysis experiments are also conducted on the smaller-scale dataset. Our experiments are based on the BELLE *3-teacher* setting. As demonstrated in Table 5, the sampling module plays a more crucial role than flux loss. Removing the flux loss results in only a slight performance degradation, whereas removing the sampling module leads to a severe performance drop. This indicates that Bayesian sampling plays a crucial role in the effectiveness of BELLE.

## 7 CONCLUSION

In this paper, we propose BELLE, a novel continuous-valued AR TTS model integrating Bayesian EDL for sampling mel-spectrogram frames. Our approach addresses the underexplored area of effective sampling methods for continuous-valued AR TTS. Additionally, a multi-teacher knowledge distillation framework is introduced, substantially improving TTS synthesis quality using synthesized data from publicly available models. Experimental results highlight the efficacy of Bayesian evidential sampling and multi-teacher distillation in achieving competitive speech naturalness, speaker similarity, and diversity, rivaling models trained on much larger real-data corpora. Remarkably, our approach attains these results using only one-tenth of the data, much of it synthetic, while enabling high-quality, low-latency TTS suitable for both offline and streaming applications.

REPRODUCIBILITY STATEMENT

To ensure reproducibility, we provide detailed model configurations, training parameters, and training data specifications in Sec. 5 and Appendix C. Upon acceptance, we will release all code, checkpoints, and synthesized speech data, enabling the community to reliably reproduce our work and further explore its potential.

ETHICS STATEMENT

BELLE introduces a novel Bayesian evidential sampling approach within continuous-valued autoregressive text-to-speech, significantly enhancing the zero-shot TTS synthesis quality. By effectively modeling and generating natural, expressive, and intelligible speech with limited reference data, BELLE advances the flexibility and realism of synthesized audio, making it valuable for various beneficial applications in society. Notably, BELLE can facilitate natural human-machine conversational systems, assistive communication technologies for individuals with speech impairments, and personalized education platforms, thereby positively impacting accessibility, education, and user experiences in interactive dialogue systems.

However, alongside the benefits, zero-shot text-to-speech technologies like BELLE also pose potential risks. They could be misused in unethical or harmful scenarios, such as impersonation, identity fraud, and targeted social engineering attacks. In principle, BELLE could mimic any person's voice from minimal audio samples, leading to malicious applications aimed at deceiving or misleading individuals. Therefore, responsible use and appropriate safeguards, such as speaker verification, synthetic audio detection, and strong regulatory oversight, are critical directions for addressing and mitigating these societal risks.

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

# A  LIMITATIONS

Although BELLE demonstrates strong performance, there are two main limitations. First, the current study focuses solely on training and evaluation with English speech data, without validating multilingual generalization; extending BELLE to diverse languages remains an important direction for future work. Second, as with most AR TTS systems, BELLE's RTF is still higher than that of recent flow-matching-based models. Further research on predicting multiple mel-spectrogram frames per step could substantially reduce RTF and enable lower-latency streaming TTS.

# B  USE OF LARGE LANGUAGE MODELS

In the preparation of this work, large language models (LLMs) were employed to refine the writing for clarity and conciseness. During the experimental phase, LLMs also provided code modification suggestions that assisted in implementing and debugging parts of the system. All conceptual designs, experimental settings, and core contributions remain the authors' original work.

# C  DETAILED EXPERIMENTAL SETUP

## C.1  TRAINING DATA

The training dataset is derived from the Librispeech (Panayotov et al., 2015) training set. We preprocess this dataset using a voice activity detection (SileroTeam, 2024) algorithm to remove prolonged silent intervals. Subsequently, to ensure quality and manageability, we pick out audio samples whose duration is from 0.5 second to 14 seconds, creating a final training dataset containing approximately 706 hours of speech. All audio samples are resampled to 16 kHz and converted into 80-dimensional mel-frequency spectrograms. Additionally, we apply grapheme-to-phoneme (G2P) [1] conversion to preprocess textual transcriptions.

To provide richer acoustic diversity necessary for robust Bayesian distribution estimation, we augment our training data by synthesizing multiple audio samples for each textual input using six publicly available pretrained TTS models, namely CosyVoice2 (Du et al., 2024) (LLM + flow matching-based streaming TTS), IndexTTS (Deng et al., 2025) (AR discrete acoustic token TTS), SparkTTS (Wang et al., 2025c) (LLM-based single-stage TTS), F5TTS (Chen et al., 2024a) (flow matching-based efficient TTS), MaskGCT (Wang et al., 2024) (masked generative NAR TTS), and XTTS-v2 (Casanova et al., 2024) (cross-lingual expressive TTS). Including the TTS-synthesized data, the total dataset amounts to 4,817 hours of speech. For consistency, all synthesized audio samples generated by these models are also resampled to 16 kHz.

## C.2  MODEL CONFIGURATIONS

Input mel-spectrograms first pass through a 3-layer prenet with dropout of 0.5 in both training and inference stages following Tacotron (Wang et al., 2017). The AR LM is a decoder-only Transformer consisting of 12 blocks, each with 16 attention heads, a hidden size of 1024, a feed-forward dimension of 4096, and a dropout rate of 0.1 [2]. The sampling module includes a linear projection to derive the sampling parameters and a 3-layer residual MLP for denoising. A post-processing module comprising five convolutional blocks (a kernel size of 5 and 256 channels) is applied for mel-spectrogram refinement. The final waveform is synthesized using the pretrained HiFi-GAN vocoder (Kong et al., 2020)[3].

## C.3  TRAINING DETAILS OF MAIN RESULTS

We train BELLE, BELLE-stream and reproduce MELLE on the data in Sec.5.1, following the training strategy of combining various data sources within each batch (see Sec. 4.4), with predefined

---

[1] https://github.com/espeak-ng/espeak-ng

[2] Our code is modified from https://github.com/lifeiteng/vall-e.

[3] The pretrained vocoder can be found in https://huggingface.co/mechanicalsea/speecht5-tts

training weights that the original Librispeech audio is weighted by 0.22 and each synthesized audio source (six TTS models) is weighted by 0.13, which ensures the original Librispeech audio receives approximately twice the weighting of synthesized audio, reflecting relative considerations of synthetic audio quality. Models are trained by AdamW optimiser with a total batch size of about 160K frames distributed across 16 NVIDIA A800 GPUs. BELLE and MELLE's training proceeds for 450K updates, where the learning rate is first linearly warmed up to a peak value of $5 \times 10^{-4}$ over the initial 10% of training steps and thereafter linearly decayed to zero. Regarding the hyperparameters for BELLE, we set $\lambda = 0.5$ in Eqn. (10) and $\lambda_{\text{samp}} = 0.2$, $\lambda_{\text{flux}} = 0.5$ in Eqn. (8). As for MELLE, our hyperparameter settings strictly follow the original MELLE paper, where $\lambda_{\text{samp}} = 0.1$, $\lambda_{\text{flux}} = 0.5$.

BELLE-stream is initialized from the trained BELLE model, trained with a batch size of 80K frames across 8 NVIDIA A800 GPUs for 150K updates, using $\lambda_{\text{flux}} = 0.1$ while keeping all other hyperparameters identical to BELLE. In our implementation, we set $S_{\text{text}} = 20$ and $S_{\text{audio}} = 50$, corresponding to an audio chunk duration of approximately 0.8 seconds. Here, $S_{\text{text}}$ is computed from the number of phonemes obtained after G2P conversion, while $S_{\text{audio}}$ is determined from the number of mel-spectrogram frames. Let $L_{\text{text}}$ and $L_{\text{audio}}$ denote the total phoneme count and the total mel-frame count of an utterance, respectively. During training, we filter out audio samples whose ratio $L_{\text{audio}} : L_{\text{text}}$ is less than 2.5, ensuring that each text chunk retains sufficient corresponding audio frames for effective streaming generation. We observed that the synthesis quality deteriorates with speech prompt. To address this, we first investigate the potential of BELLE for streaming synthesis in a single-speaker scenario, finetuning BELLE-stream using recordings from a single speaker of Librispeech to fix its voice timbre.

## C.4 TRAINING DETAILS OF ANALYSIS RESULTS

Due to limited computational resources, the analysis experiments are conducted on a smaller-scale dataset. The training dataset is derived from the Librispeech train-clean 100-hour subset, following the same processing procedure in Sec. 5.1, which contains approximately 72 hours of speech. Training solely on the original Librispeech data without synthesized augmentation is denoted as *1-teacher*. The *3-teacher* condition introduces additional synthesized speech from XTTS-v2 and MaskGCT, about 218h. The *7-teacher* condition includes augmented speech from all six TTS models mentioned in Sec. 5.1, about 455h. The *data-aug* configuration serves as a strong baseline using a conventional data augmentation strategy, where data from all sources totaling 455 hours is randomly sampled during training. The *3-teacher* and *7-teacher* follow the training strategy in Sec. 4.4, with a total batch size of about 80K frames and 47K training steps.

## C.5 EVALUATION SETTINGS

We evaluate the zero-shot TTS capabilities of our model using the LibriSpeech test-clean subset following VALL-E (Wang et al., 2023) by an open-source evaluation protocol (Lee, 2024). Specifically, we consider two inference conditions: **Continuation**, in which the first 3 seconds of an utterance and its corresponding transcription serve as the prompt, and the model synthesizes the continuation of speech thereafter; and **Cross-sentence**, where we use a reference utterance and its corresponding transcription from a given speaker as a prompt, and then the model generates speech for a different sentence while preserving speaker characteristics.

For objective evaluation, we report word error rate (WER) to measure intelligibility and robustness, using two automatic speech recognition models: a Conformer-Transducer[4] (Gulati et al., 2020) and a fine-tuned HuBERT-Large model[5] (Hsu et al., 2021). We denote results obtained from these systems as WER-C and WER-H, respectively. To quantify speaker similarity, we calculate the cosine similarity between extracted speaker embeddings using a WavLM-TDCNN model[6] (Chen et al., 2022). We provide two similarity metrics: SIM-o computes the similarity against the original speech prompt, whereas SIM-r uses the vocoder-reconstructed version of the prompt.

---

[4]`https://huggingface.co/nvidia/stt_en_conformer_transducer_xlarge`

[5]`https://huggingface.co/facebook/hubert-large-ls960-ft`

[6]`https://github.com/microsoft/UniSpeech/tree/main/downstreams/speaker_verification`

For subjective evaluation, we obtain MOS and SMOS scores via a crowdsourcing platform. MOS for evaluating overall speech quality, and SMOS for measuring speaker similarity between the generated audio and the prompt. From the audio samples generated under the Cross-sentence setting, we select one sample per speaker, resulting in a total of 40 audio samples. All samples from different systems are resampled to 16kHz to ensure a fair comparison. We evaluate MOS and SMOS following the detailed procedure described in Appendix F. Note that BELLE-stream is fine-tuned to a fixed speaker timbre and does not use an audio prompt; therefore, it is not evaluated for SMOS or SIM metrics, and its results are reported only under the Cross-sentence setting.

For diversity evaluation, building on the layer-wise analysis of WavLM in (Chiu et al., 2025), we adopt the 13th (of 24) layer of WavLM-Large as the speaker representation; 1,234 audio prompts are sampled from the LibriSpeech test-clean subset, and under the Cross-sentence setting each prompt is inferred three times to obtain 1,234 groups of outputs (three per group); frame-level hidden states from layer 13 are extracted for each generation, mean-pooled and L2-normalized, and within-group pairwise cosine similarity, L1, and L2 distances are computed; corpus-level means and standard deviations across all 1,234 groups are reported to quantify the diversity of the generated speech.

To maintain fairness and consistency across all evaluations, we filter the test utterances to only those between 4 and 10 seconds in duration, and report all evaluation metrics using a fixed evaluation set shared among all compared models.

## D DETAILS ABOUT BAYESIAN EVIDENTIAL LEARNING

Bayesian Evidential Learning (EDL) provides a principled framework for uncertainty estimation in regression tasks by explicitly modeling the posterior distribution of predictions. Observed data $y$ are assumed to be drawn independently and identically distributed (i.i.d.) from a Gaussian distribution with unknown mean and variance. According to Maximum Likelihood Estimation (MLE), we aim to find parameters $\mu$ and $\sigma^2$ that maximize the likelihood of observing the given data $y$, or equivalently, minimize the Negative Log Likelihood (NLL) loss. We model this problem by placing prior distributions on both the mean and variance:

$$y_1, \ldots, y_N \sim \mathcal{N}(\mu, \sigma^2), \quad \mu \sim \mathcal{N}\left(\gamma, \frac{\sigma^2}{\nu}\right), \quad \sigma^2 \sim \Gamma^{-1}(\alpha, \beta) \tag{13}$$

where $\Gamma(\cdot)$ denotes the gamma function, $\boldsymbol{m} = (\gamma, \nu, \alpha, \beta)$ summarizes the set of hyperparameters, and the constraints are given by $\gamma \in \mathbb{R}$, $\nu > 0$, $\alpha > 1$, and $\beta > 0$. We set $q(\mu, \sigma^2) = p(\mu, \sigma^2 \mid y_1, \ldots, y_N)$ as our target posterior. For tractability, we adopt a approximation, assuming $q(\mu, \sigma^2) = q(\mu)q(\sigma^2)$. Under this assumption, we use the Normal Inverse-Gamma (NIG) distribution as our conjugate prior, which can be written as:

$$p(\mu, \sigma^2 \mid \gamma, \nu, \alpha, \beta) = \frac{\beta^\alpha \sqrt{\nu}}{\Gamma(\alpha)\sqrt{2\pi\sigma^2}} \left(\frac{1}{\sigma^2}\right)^{\alpha+1} \exp\left\{-\frac{2\beta + \nu(\gamma - \mu)^2}{2\sigma^2}\right\}. \tag{14}$$

Denote $\boldsymbol{\theta} = (\mu, \sigma^2)$, marginalizing out $\mu$ and $\sigma^2$:

$$p(y_i \mid \boldsymbol{m}) = \frac{p(y_i \mid \boldsymbol{\theta}, \boldsymbol{m})p(\boldsymbol{\theta} \mid \boldsymbol{m})}{p(\boldsymbol{\theta} \mid y_i, \boldsymbol{m})} = \int_{\sigma^2=0}^{\infty} \int_{\mu=-\infty}^{\infty} p(y_i \mid \mu, \sigma^2)\, p(\mu, \sigma^2 \mid \boldsymbol{m})\, \mathrm{d}\mu \, \mathrm{d}\sigma^2 \tag{15}$$

Substituting the Gaussian likelihood and the Normal Inverse-Gamma prior into the above equation yields an analytically tractable form, Student-$t$ distribution:

$$p(y_i | \gamma, \nu, \alpha, \beta) = \mathrm{St}\left(y_i; \gamma, \frac{\beta(1+\nu)}{\nu\alpha}, 2\alpha\right), \tag{16}$$

where $\gamma$ is the location, $\frac{\beta(1+\nu)}{\nu\alpha}$ is the scale, and $2\alpha$ is the degrees of freedom.

In practice, a neural network predicts the NIG parameters directly. The training loss, termed *evidential loss*, comprises a negative log-likelihood term and a regularization term that penalizes incorrect evidence:

$$\mathcal{L}_{\mathrm{EDL}}(y_i) = \mathcal{L}_{\mathrm{NLL}}(y_i) + \lambda \mathcal{L}_{\mathrm{R}}(y_i) \tag{17}$$

where

$$\mathcal{L}_{\text{NLL}}(y_i) = \frac{1}{2}\log\left(\frac{\pi}{\nu}\right) - \alpha\log(\Omega) + \left(\alpha + \frac{1}{2}\right)\log\left(\nu(y_i - \gamma)^2 + \Omega\right) + \log\left(\frac{\Gamma(\alpha)}{\Gamma\left(\alpha + \frac{1}{2}\right)}\right) \quad (18)$$

$$\mathcal{L}_{\text{R}}(y_i) = |y_i - \gamma| \cdot (2\nu + \alpha) \quad (19)$$

with $\Omega = 2\beta(1 + \nu)$.

## E  MELLE

### E.1  GAUSSIAN SAMPLING IN MELLE

In MELLE (Meng et al., 2024), the latent sampling module assumes that the embedding at timestep $t$, denoted as $\boldsymbol{z}_t$, is drawn from a multivariate Gaussian distribution:

$$\boldsymbol{z}_t \sim \mathcal{N}(\boldsymbol{\mu}_t, \boldsymbol{\sigma}_t^2 \boldsymbol{I}), \quad (20)$$

where the mean $\boldsymbol{\mu}_t \in \mathbb{R}^D$ and the log-variance $\log\boldsymbol{\sigma}_t^2 \in \mathbb{R}^D$, with $D$ representing the number of mel-frequency bands, are computed from the autoregressive hidden representation $\boldsymbol{e}_t$ via a linear transformation:

$$\left[\boldsymbol{\mu}_t, \log\boldsymbol{\sigma}_t^2\right] = \boldsymbol{W}_{\text{G}}\boldsymbol{e}_t + \boldsymbol{b}_{\text{G}}. \quad (21)$$

Sampling is performed using the standard reparameterization trick:

$$\boldsymbol{z}_t = \boldsymbol{\mu}_t + \boldsymbol{\sigma}_t \odot \boldsymbol{\epsilon}, \quad \boldsymbol{\epsilon} \sim \mathcal{N}(\boldsymbol{0}, \boldsymbol{I}). \quad (22)$$

### E.2  SAMPLING LOSS

When Gaussian sampling is used, the sampling loss is typically implemented as a Kullback–Leibler (KL) divergence loss between the predicted Gaussian distribution and a predefined Gaussian prior distribution with mean $\boldsymbol{y}^{\text{gt}}$ and variance $\boldsymbol{I}$.

## F  SUBJECTIVE EVALUATION

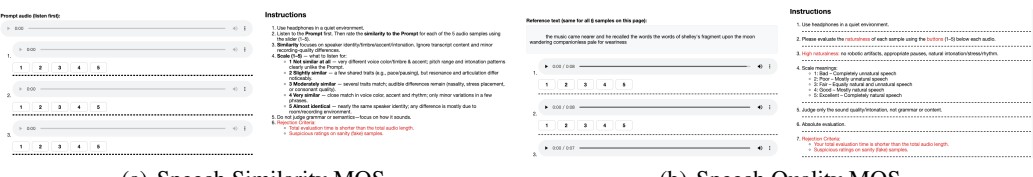

(a) Speech Similarity MOS        (b) Speech Quality MOS

Figure 2: Screenshots of subjective evaluations.

For the TTS task, we focus on the Mean Opinion Score (MOS) and Speaker Similarity (SMOS). From the audio samples generated under the Cross-sentence setting, we select one sample per speaker, resulting in a total of 40 audio samples. All samples from different systems are resampled to 16kHz to ensure a fair comparison. The details are as follows: For speech quality evaluation, we conducted an MOS (Mean Opinion Score) test and explicitly instructed the raters to focus on assessing audio quality and naturalness, while ignoring differences in style (e.g., timbre, emotion, and prosody). The raters were presented with and scored samples, and each rater was asked to evaluate the subjective naturalness on a 1-5 Likert scale. When scoring, the order of audio samples is randomly shuffled.

For speaker similarity evaluation, we asked the raters to focus on the similarity of the speaker's identity (timbre) to the reference, while ignoring differences in content, grammar, or audio quality. We paired each synthetic utterance with a reference utterance to assess the degree of matching between the synthesized speech and the target speaker. Each rater was asked to evaluate the speaker similarity on a 1-5 Likert scale. When scoring, the order of audio samples is randomly shuffled.

Our subjective evaluation was crowd-sourced, with 15 native speakers participating via Amazon Mechanical Turk. The instructions for the testers are shown in the Figure 2. We paid approximately $100 in participant compensation. A small portion of the speech samples used in the test can be heard at the following website: https://belletts.github.io/Belle/.

