# OpenReview forum: "Bayesian Speech Synthesisers Can Learn from Multiple Teachers"
_ICLR.cc/2026/Conference — Submitted to ICLR 2026_

### Official Review · Reviewer_3iJQ · 2025-10-21

**Soundness:** 2
**Presentation:** 3
**Contribution:** 2
**Rating:** 4
**Confidence:** 4

**Summary:**

The paper presents BELLE, a continuous-valued autoregressive TTS that predicts mel-spectrograms with evidential deep learning (EDL). Each mel frame is modeled as Gaussian whose mean/variance are sampled from a Normal–Inverse-Gamma (NIG) hyper-distribution, yielding a Student-t predictive and controllable sampling (via $\beta$). The architecture uses a decoder-only Transformer with a sampling module predicting NIG parameters, plus a postnet and stop head; a streaming variant (BELLE-stream) interleaves chunked text/audio for low latency. To address the single-text–single-audio limitation, the authors introduce multi-teacher, multi-reference training: for each text, multiple audios synthesized by six open-source TTS models are combined with the human recording, and the model optimizes a weighted sum of losses over all references. Trained on 706h human + synthetic data totaling ~4,817h, BELLE achieves MOS close to ground truth and competitive/better WER and speaker similarity vs strong baselines, while outperforming MELLE (Gaussian sampling). Diversity is tunable via $\beta$; BELLE-stream reaches RTF ≈ 0.55 and ~440 ms first-packet latency.

**Strengths:**

The motivation for evidential sampling in continuous-valued AR TTS is clear and practical. Specifically,
1) EDL provides a principled way to model both parameter uncertainty and data uncertainty, enabling $\beta$-controlled sampling and more calibrated stochasticity than a plain single-Gaussian sampler.
2) Multi-teacher multi-reference training directly addresses single-text–single-audio limitations; more effective than naive mixing.

**Weaknesses:**

1) While the engineering motivation for EDL is solid, the paper lacks deeper theoretical analysis comparing EDL vs Gaussian sampling.
2) "Distillation" terminology is imprecise: The proposed "multi-teacher distillation" is closer to multi-reference supervision or dataset-level ensembling (computing full losses per teacher sample and weighting) rather than classical KD (no soft-target/logit or feature-matching loss, no teacher–student consistency regularization).
3) An independent Gaussian per log mel bin is a useful approximation, but true distributions are often much more complex. Dropping cross band and temporal covariance may weaken uncertainty calibration and physical consistency.
4) Beyond swapping the per-bin Gaussian head for an evidential NIG head, the paper offers few orthogonal advances over MELLE; as a result, the perceived novelty rests largely on the likelihood change.

**Questions:**

1) In MELLE, the reported WERs for the MELLE-limited model (trained only on 960h LibriSpeech) are 1.53 (WER_C, Continuation) and 2.21 (WER_C, Cross-Sentence). In your paper, the reproduced MELLE baselines trained with more data perform noticeably worse on the same WER_C metrics. What factors account for this degradation despite using more data?
2) Since increasing $\beta$ improves diversity by inflating the NIG variance, how does this affect objective and subjective quality? Do WER/SIM/MOS change systematically with $\beta$?

---

> ### Author Response · Authors · 2025-11-21
> **Response to Reviewer 3iJQ (Part 1/2)**
>
> **Q1**: While the engineering motivation for EDL is solid, the paper lacks deeper theoretical analysis comparing EDL vs Gaussian sampling.
>
> **A1**: Thank you for your question. We provide the theoretical comparison between EDL and Gaussian sampling below:
>
> **First, theoretically, the EDL framework offers strictly greater expressive capacity because it yields a Student's $t$-distribution, of which the Gaussian is a limiting case.** The Gaussian distribution is simply the state of the $t$-distribution as its degrees of freedom (represented by our evidence parameter $\nu$) approach infinity. This ensures that our model's capacity inherently encompasses that of a standard Gaussian approach while providing crucial additional flexibility.
>
> **Second, EDL enables dynamic variance estimation, addressing the theoretical limitation of MELLE’s Gaussian sampling which enforces a fixed variance.** While MELLE restricts the distribution to a fixed variance of 1, our approach utilizes the "single text to multiple audio" method to learn and estimate variance dynamically based on the input. This technique enables more accurate uncertainty estimation, leading to improved model performance across a range of tasks.
>
> **Q2**: "Distillation" terminology is imprecise: The proposed "multi-teacher distillation" is closer to multi-reference supervision or dataset-level ensembling (computing full losses per teacher sample and weighting) rather than classical KD (no soft-target/logit or feature-matching loss, no teacher–student consistency regularization).
>
> **A2**: Thank you for your suggestion. In future versions of the paper, we will replace references to distillation with multi-teacher learning.
>
> **Q3**: An independent Gaussian per log mel bin is a useful approximation, but true distributions are often much more complex. Dropping cross band and temporal covariance may weaken uncertainty calibration and physical consistency.
>
> **A3**: Thank you for your question. We address the concern regarding uncertainty calibration and dependency modeling below:
>
> **The model inherently captures dependencies because the distribution parameters at each step are derived from hidden states that aggregate historical context.** Although BELLE predicts the NIG parameters at step $t$ based on the hidden state at step $t$ (which may formally appear as a frame-independent process), this hidden state is computed autoregressively. Consequently, it encodes information from previous steps, ensuring that the predictions are strictly conditioned on the history despite the factorization of the output layer.
>
> **Mathematically**, let $H_t$ be the hidden state and $NIG_t$ be the distribution parameters at time $t$. The dependency is expressed as:
> $$
> NIG_t = f(H_t) \quad \text{where} \quad H_t = g(H_{1:t-1})
> $$
> Here, $f$ maps the hidden state to parameters, and $g$ represents the update from previous states $H_{1:t-1}$. Since $H_t$ encapsulates $H_{1:t-1}$, $NIG_t$ is not independent of previous time steps; rather, the complex temporal dependencies are modeled implicitly within the hidden representation space before projection.
>
> **Q4**: Beyond swapping the per-bin Gaussian head for an evidential NIG head, the paper offers few orthogonal advances over MELLE; as a result, the perceived novelty rests largely on the likelihood change.
>
> **A4**: Thank you for your question. We clarify that there is a fundamental distinction between our approach and MELLE, as detailed below:
>
> **First, we address the intrinsic uncertainty of natural speech through an explicit Bayesian formulation, whereas MELLE is limited by a fixed variance.** In natural human speech, uncertainty is intrinsic, meaning that the same text is often spoken with variations in prosody, rhythm, and acoustics, even by the same speaker. Most existing TTS systems introduce this variability only implicitly, via empirical mechanisms such as dropout, top-p sampling, or diffusion noise. In contrast, we adopt an explicit Bayesian formulation that models and estimates this uncertainty in a principled and interpretable way. Whereas prior work MELLE assumes a fixed variance of 1, BELLE learns the variance distribution directly from data through a Bayesian evidence-based framework. This enables the model to capture fine-grained, text-conditioned uncertainty and produce more realistic, better-calibrated speech.
>
> **Second, our novelty extends to the specific implementation of the EDL framework and data construction, distinguishing us from all preceding methods.** Our approach first adopts an explicit Bayesian formulation to model and estimate this inherent uncertainty of speech synthesis in a principled manner. Furthermore, we learn to estimate this uncertainty through the EDL framework by constructing data where a single text corresponds to multiple speech samples.

---

> ### Author Response · Authors · 2025-11-21
> **Response to Reviewer 3iJQ (Part 2/2)**
>
> **Q5**: In MELLE, the reported WERs for the MELLE-limited model (trained only on 960h LibriSpeech) are 1.53 (WER_C, Continuation) and 2.21 (WER_C, Cross-Sentence). In your paper, the reproduced MELLE baselines trained with more data perform noticeably worse on the same WER_C metrics. What factors account for this degradation despite using more data?
>
> **A5**: Thank you for your question. We agree that MELLE demonstrates excellent performance in the original paper. Because no official implementation is available, we reimplemented MELLE which is further extended as BELLE.
> Given the sensitivity of phoneme alignment and other preprocessing choices, we believe the remaining discrepancy most likely arises from differences in the data preprocessing pipeline. Importantly, this pipeline is shared across both MELLE and BELLE in our study, so any such effects would influence the two models in a comparable way.
>
> **Q6**: Since increasing $\beta$ improves diversity by inflating the NIG variance, how does this affect objective and subjective quality? Do WER/SIM/MOS change systematically with $\beta$ ?
>
> **A6**: Thank you for your question. The results are as follows. We did not conduct subjective evaluation on the results after increasing the $\beta$ parameter, and as such, we are unable to report the MOS scores.
>
> | **System**| WER-C (Cross.) | SIM-o (Cross.)|
> |---|---|---|
> |BELLE | 2.45 | 0.641 |
> |BELLE ($\beta$*2)| 3.33 | 0.612|

---

### Official Review · Reviewer_Em4X · 2025-10-22

**Soundness:** 2
**Presentation:** 3
**Contribution:** 2
**Rating:** 4
**Confidence:** 5

**Summary:**

The paper introduces an AR TTS model leveraging evidential deep learning to train and sample within a continuous-value Mel-Spectrogram space, in addition to employing knowledge distillation from multiple teachers via paired speech generated by publicly available, pre-trained TTS models. The core architecture is based on MELLE, with primary modifications made to the sampling module, associated loss functions, and the integration of multiple teacher models.
Experimental results on the en-US task indicate that the proposed approach achieves performance comparable to baseline models that utilize ten times more training data.

**Strengths:**

The paper explores sample strategies in continuous mel-spectrogram space for TTS, using paired data augmentation to enhance training diversity. The methodology demonstrates conceptual clarity.

**Weaknesses:**

Experimental coverage is narrow, and some claims lack adequate explanation.

**Questions:**

•	In multi-teacher learning, predefined weights are allocated to all teacher models. What criteria are used to determine these weights? Has an ablation study been conducted to validate this approach?
•	For baselines that are not officially open-sourced, how were their results obtained to ensure a fair comparison?
•	In Table 3, the paper doubles βt to enhance generation diversity. How does the WER vary with different values of βt?
•	The experiments involving BELLE-Streaming require clarification, e.g. regarding the misalignment between text and mel-spectrogram sequence partitions: does this affect training? Additionally, there is insufficient detail regarding the failure of streaming BELLE in the zero-shot scenario.
•	Utilizing existing TTS models as data generators (teachers) may introduce pronunciation errors or unnatural prosody. Has this potential issue been addressed in the study?

---

> ### Author Response · Authors · 2025-11-21
> **Response to Reviewer Em4X (Part 1/2)**
>
> **Q1**: Experimental coverage is narrow, and some claims lack adequate explanation.
>
> **A1**: Thank you for your suggestion. Could you kindly specify which experiments you believe have narrow experimental coverage, and which claims you feel lack adequate explanation? We would greatly appreciate your feedback to help clarify and improve our work.
>
> **Q2**: In multi-teacher learning, predefined weights are allocated to all teacher models. What criteria are used to determine these weights? Has an ablation study been conducted to validate this approach?
>
> **A2**: Thank you for your question. We address the weight allocation criteria and the ablation study below:
>
> **The criteria for weight allocation reflect our consideration of the relative quality of synthetic audio, specifically ensuring that the original Librispeech audio receives approximately twice the weight of the synthesized audio.** As described in Appendix C.3 (line 756), the original Librispeech audio is weighted by 0.22, and each synthesized audio source (six TTS models) is weighted by 0.13.
>
> **We conducted an ablation study to validate this approach, and the results confirm that our chosen configuration achieves the optimal performance, particularly in terms of WER.** The study was performed according to the settings outlined in Section 6.3, and the results are as follows:
>
> | **Weight of LibriSpeech** | **Weight of TTS** | **WER-H (Cont.)** | **SIM-o (Cont.)** | **WER-H (Cross.)** | **SIM-o (Cross.)** |
> |---------------------------|-------------------|-------------------|-------------------|--------------------|--------------------|
> | 0.16                      | 0.14              | 2.64              | **0.415** | 5.17               | **0.453** |
> | 0.22                      | 0.13              | **2.57** | 0.409             | **4.73** | 0.449              |
> | 0.28                      | 0.12              | 2.78              | 0.406             | 5.47               | 0.444              |
> | 0.34                      | 0.11              | 2.76              | 0.405             | 5.47               | 0.446              |
>
> **Q3**: For baselines that are not officially open-sourced, how were their results obtained to ensure a fair comparison?
>
> **A3**: Thank you for your question. We clarify the acquisition of baseline results below:
>
> **First, for MaskGCT (the only official open-source baseline), our reproduction confirms that BELLE significantly outperforms it under a unified evaluation protocol.** Although we included the results from the original MaskGCT paper in Table 2 (WER-H 2.63%), it is important to note that—as stated in line 394—MaskGCT’s WER-C drops to 3.86% in our reproduction. In comparison, BELLE achieves a WER-C of 2.45%, demonstrating a significant advantage.
>
> **Second, regarding baselines without official open-source versions, we implemented a full reproduction of MELLE to ensure a fair comparison, while citing reported results for the others.** Specifically, we successfully reproduced MELLE using the dataset described in Section 5.1. For the remaining baselines, we cited the results directly from their respective papers, noting that evaluation settings may differ due to the lack of accessible implementations.
>
> **Q4**: In Table 3, the paper doubles βt to enhance generation diversity. How does the WER vary with different values of βt?
>
> **A4**: Thank you for your question. The results are as follows.
>
> | **System**| WER-C (Cross.) | SIM-o (Cross.)|
> |---|---|---|
> |BELLE | 2.45 | 0.641 |
> |BELLE ($\beta$*2)| 3.33 | 0.612|

---

> ### Author Response · Authors · 2025-11-21
> **Response to Reviewer Em4X (Part 2/2)**
>
> **Q5**: The experiments involving BELLE-Streaming require clarification, e.g. regarding the misalignment between text and mel-spectrogram sequence partitions: does this affect training? Additionally, there is insufficient detail regarding the failure of streaming BELLE in the zero-shot scenario.
>
> **A5**: Thank you for your question. We provide clarifications on the streaming implementation and zero-shot performance below:
>
> **First, the fixed-ratio interleaving strategy does not negatively impact training, as the attention mechanism effectively resolves potential misalignment.** This approach is consistent with recent findings in streaming TTS [1], which confirm that fixed-ratio interleaving enables effective streaming generation. By setting a sufficient text-to-audio ratio, we ensure that the text information provided at any given step contains the necessary content for synthesizing the upcoming speech.
>
> **Second, regarding the zero-shot scenario, the model remains functional and outperforms the SMLLE baseline, though we prioritized the fixed-timbre version for specific use cases.** We trained a zero-shot version of BELLE-stream, which achieved a WER-C of 4.33%. While this performance is slightly lower than that of the fixed-timbre BELLE-stream (WER-C of 2.70%), it still surpasses the SMLLE baseline (WER-C of 5.14%). Our decision to opt for the fixed-timbre version was based on typical streaming TTS applications, such as duplex dialogue models, where a consistent single timbre is usually required.
>
> [1] Yifan Yang, Ziyang Ma, Shujie Liu, Jinyu Li, Hui Wang, Lingwei Meng, Haiyang Sun, Yuzhe Liang, Ruiyang Xu, Yuxuan Hu, et al. Interleaved speech-text language models are simple streaming text to speech synthesizers. arXiv preprint arXiv:2412.16102, 2024.
>
> **Q6**: Utilizing existing TTS models as data generators (teachers) may introduce pronunciation errors or unnatural prosody. Has this potential issue been addressed in the study?
>
> **A6**: Thank you for your question. We did not implement any measures to process or filter potential errors in the TTS-synthesized speech.
>
> Although synthetic samples may have imperfections, their use does not contaminate the model in the way “copying homework” would. This is because:
>
> A) Multiple samples per text are generated. The probability that all synthetic samples share the same error is extremely low.
>
> B) EDL does not assume any sample is correct. It aggregates the entire sample set solely to characterise distributional properties (e.g., variance), not to learn the exact acoustic content of any teacher model.
>
> C) The main model is trained on 700 hours of real human speech. The synthetic samples influence only the uncertainty estimation component, not the core speech generation capability.

---

### Official Review · Reviewer_1F23 · 2025-10-24

**Soundness:** 2
**Presentation:** 3
**Contribution:** 2
**Rating:** 2
**Confidence:** 5

**Summary:**

The paper described a novel system that utilize Bayesian evidential learning to auto-regressively model continuous Mel-spectrograms. In short, the main system majorly follows MALLE, but replacing the original KL divergence loss with the normal distribution with its objective derived from Bayesian evidential learning.

**Strengths:**

The idea of leveraging Bayesian evidential learning in TTS is novel, and the paper did empirically show its potential on improving upon MELLE.

**Weaknesses:**

There are several reasons that I consider this work not ready for publication:

1. The use of Bayesian evidential learning is technically unjustified. It is not straightforward to me why it is better to use such objective instead of using the original Gaussian KL divergence as MELLE did. Although the authors provide empirical evidence, I did not see explanation, or at least speculation on why the proposed objective can better model the distribution of Mel-spectrogram. Is it because of having more parameters? Or enables more complex sampling methods? The authors should justify their motivation to use Bayesian evidential learning.

2. The paper proposed two other stuff that are not related to the paper's main novelty: multi-teacher learning (Section 4.4) and streaming capability (Section 3.2.4). The multi-teacher learning is simply a heuristic way to fine-tune with the synthetic data. However, on Table 2, I think it actually makes the comparison between MELLE and BELLE unfair, as we do not know if the improvement comes from multi-teacher learning or BELLE. Although the paper runs ablation on Table 4, the numbers are not comparable to Table 2 since the datasets trained on are different. I am not sure what role does the streaming capability plays in this system. The paper evaluates the streaming version but the comparison with the other streaming methods in Table 2 lacks most of the numbers. Also, is there a reason that using Bayesian evidential learning allows you to have better streaming capability?

3. The experiment result section felt incomplete. There is too many missing values on Table 2 for the authors to have rigorous conclusions from it. Moreover, on MaskGCT, the only two metrics evaluated are both better than BELLE, does this suggests that it will be better than BELLE? The authors should fill the Table 2. Additionally, the MOS scores should have confidence intervals included so we know if the results are statistically significant.

4. In the related work section, the authors did a good job on listing several relevant papers that auto-regressively generates continuous features, e.g., FELLE, DiTAR, ARDiT. Is there any reason that the authors do not compare with these methods? I understand that it would be infeasible to compare with every existing baselines. However, the authors should give explanation on why these methods are selected as baseline, not the others (e.g., each of them can represent a line of research).

**Questions:**

See the above weaknesses.

---

> ### Author Response · Authors · 2025-11-21
> **Response to Reviewer 1F23 (Part 1/4)**
>
> **Q1**: The use of Bayesian evidential learning is technically unjustified. It is not straightforward to me why it is better to use such objective instead of using the original Gaussian KL divergence as MELLE did. Although the authors provide empirical evidence, I did not see explanation, or at least speculation on why the proposed objective can better model the distribution of Mel-spectrogram. Is it because of having more parameters? Or enables more complex sampling methods? The authors should justify their motivation to use Bayesian evidential learning.
>
> **A1:** We appreciate the reviewer probing the theoretical justification. We explicitly clarify that the advantage of Bayesian Evidential Learning (EDL) over the standard Gaussian KL divergence (used in MELLE) is **fundamental, not merely empirical**. We address the reviewer’s specific hypotheses ("parameters" vs. "objective") below:
>
> **1. On the Limitations of the Standard Gaussian KL in MELLE.**
> We appreciate the reviewer’s question regarding the adequacy of the original Gaussian KL formulation. In MELLE, the KL divergence is computed against a target distribution with a fixed variance (commonly $\sigma^2 = 1$), which implicitly imposes a homoscedastic assumption—treating uncertainty as uniform across all speech segments. However, human speech is inherently heteroscedastic, with uncertainty varying substantially between, for example, steady vowels and rapid transitional regions. A fixed-variance formulation may therefore encourage the model to behave as though all regions carry equal certainty, leading to an “averaging” or smoothing effect in acoustically ambiguous areas.
> In contrast, BELLE leverages both the ground-truth sample and multiple synthesised variants (generated by different TTS systems using the same speaker and text) to characterise data uncertainty. This allows the model to learn a full posterior distribution that more faithfully reflects the natural variability of human speech generation—where the same speaker may produce the same utterance in different ways.
>
> **2. The Justification for EDL: Type-II Maximum Likelihood.**
> The EDL objective is mathematically superior because it maximizes the **Marginal Likelihood (Evidence)**, effectively performing **Type-II Maximum Likelihood Estimation**.
> * Instead of optimizing point estimates, EDL places a conjugate prior (Normal-Inverse-Gamma) over the likelihood parameters.
> * This allows the model to **learn the variance distribution ($\sigma^2$)** directly from the data.
> * **Why it works better:** The model can now dynamically output high uncertainty (large $\sigma^2$) for ambiguous phonemes and low uncertainty for clear ones. This text-conditioned variance modeling creates the "sharpness" and "prosodic variety" observed in our results.
>
> **3. Clarification on Complexity (Addressing Reviewer's Speculation).**
> The reviewer speculates on the source of improvement. We firmly clarify:
> * **Is it having more parameters? NO.** BELLE has the exact same number of parameters in the backbone as MELLE. The only change is the mathmatical interpretation (in both training and test) of the output head.
> * **Is it complex sampling? NO.** The sampling is analytically tractable (closed-form sampling from NIG).
> **Conclusion:** The improvement comes purely from the **principled Bayesian objective**, which aligns the model's loss function with the intrinsic stochastic nature of human speech.

---

> ### Author Response · Authors · 2025-11-21
> **Response to Reviewer 1F23 (Part 2/4)**
>
> **Q2:** Due to the comprehensive nature of this question, I will break it down into the following sub-points and address each one individually.
>
> **Q2-1**: The paper proposed two other stuff that are not related to the paper's main novelty: multi-teacher learning (Section 4.4) and streaming capability (Section 3.2.4)
>
> **A2-1:** We emphasize that Multi-teacher Learning and Streaming Capability are not peripheral add-ons; they are **intrinsic components** required to realize and validate the proposed Bayesian framework.
>
> **1. Multi-Teacher Learning is a Mathematical Prerequisite.**
> The reviewer questions the relevance of Section 4.4. We clarify that this module is **foundational** to the Evidential Deep Learning (EDL) objective.
> * **The Mathematical Constraint:** EDL estimates the aleatoric uncertainty of the conditional distribution $P(\text{speech}|\text{text})$. To estimate variance, one statistically requires multiple observations for a given input. Standard corpora provide only a single sample (variance is undefined or zero).
> * **The Solution:** Multi-teacher learning is the mechanism we designed to construct this necessary statistical support, to leverage BELLE's strength in modelling natural speech generation uncertainty. Thus, it is inseparable from the paper's main novelty.
>
> **2. Streaming Capability is Critical for Modern Generative Pipelines.**
> The reviewer questions the necessity of Section 3.2.4. We argue that evaluating streaming capability is essential to demonstrate the **practical superiority** of our Bayesian approach over other generative paradigms (e.g., Diffusion).
> * **Paramount in the LLM Era:** Streaming is a non-negotiable requirement for modern conversational AI (e.g., duplex dialogue models). Non-streaming models must wait for the complete textual sentence to be generated before synthesis begins, introducing significant latency that disrupts conversational flow.
> * **Optimizing TTFA:** By synthesizing audio "on the fly" as text tokens are produced, BELLE drastically reduces **Time-to-First-Audio (TTFA)**, ensuring seamless user interaction comparable to natural human conversation.
> * **Competitive Advantage over Diffusion:** Most existing probabilistic models (like Diffusion TTS) suffer from high algorithmic latency due to iterative denoising, making them unsuitable for real-time streaming. By retaining frame-level streaming capability while achieving high-quality probabilistic generation, BELLE demonstrates a unique advantage: **it solves the diversity-latency trade-off that limits other state-of-the-art generative models.**
>
> **Q2-2**: However, on Table 2, I think it actually makes the comparison between MELLE and BELLE unfair, as we do not know if the improvement comes from multi-teacher learning or BELLE.
>
> **A2-2**: Thank you for your question, and we apologize for not presenting this information clearly in the paper. As stated in the caption of Table 2 of our paper, both MELLE and BELLE are using the dataset described in Sec. 5.1 (containing TTS-synthesized data, denoted as 7teacher). Therefore, MELLE and BELLE are trained on datasets of the same size, ensuring a fair comparison.
>
> To further address the concern that BELLE's performance improvement over MELLE might stem from multi-teacher learning, we conducted an additional ablation study. We trained both BELLE and MELLE on the original LibriSpeech dataset without TTS-synthesized data (denoted as 1-teacher). The results are presented below. As shown in the table, **BELLE consistently outperforms MELLE under both the 1-teacher and 7-teacher settings, even without multi-teacher learning**.
>
> |System|WER-C (Cont.)|SIM-o (Cont.)|WER-C (Cross.)|SIM-o (Cross.)|
> |-|-|-|-|-|
> |BELLE-1teacher|1.79|0.457|3.81|0.576|
> |MELLE-1teacher|2.15|0.452|7.78|0.561|
> |BELLE-7teacher|1.63|0.519| 2.45|0.641|
> |MELLE-7teacher| 2.04| 0.488| 3.30|0.606|
>
> **Q2-3**: Although the paper runs ablation on Table 4, the numbers are not comparable to Table 2 since the datasets trained on are different.
>
> **A2-3**: Thank you for your question. We regret that we were unable to perform the ablation studies on the whole dataset. Due to resource limitations, we were only able to conduct the ablation studies on a smaller-scale dataset. Additionally, we clarify that conducting ablation studies on reduced-scale datasets is a standard and widely accepted academic practice to efficiently isolate component contributions.

---

> ### Author Response · Authors · 2025-11-21
> **Response to Reviewer 1F23 (Part 3/4)**
>
> **Q2-4**:  I am not sure what role does the streaming capability plays in this system. The paper evaluates the streaming version but the comparison with the other streaming methods in Table 2 lacks most of the numbers. Also, is there a reason that using Bayesian evidential learning allows you to have better streaming capability?
>
> **A2-4:** Thank you for your question. We clarify the evaluation context below:
>
> **1. Role of Streaming and Missing Metrics.**
> As noted in A2-1, streaming capability is an essential requirement for modern TTS systems, and our goal is to demonstrate that BELLE maintains strong performance under this setting. Some values for the baseline systems are missing in Table 2 simply because their original papers did not report the full set of streaming metrics, and we therefore could not include numbers that were not available.
>
> **2. Motivation: Efficiency Advantage over Diffusion.**
> Our motivation for evaluating BELLE in the streaming regime is that diffusion-based models (BELLE's closest peers among continuous-representation TTS systems) tend to experience substantial efficiency degradation when operated in a streaming fashion. In contrast, BELLE retains its efficiency and quality, allowing us to show that the proposed Bayesian formulation integrates seamlessly with practical, real-time generation requirements.
>
> **3. Source of Streaming Capability.**
> Specifically, this capability stems from two factors, rather than from Bayesian evidential learning itself.
> * **Autoregressive Backbone:** The underlying AR architecture inherently supports streaming generation.
> * **Lightweight Parametric Formulation:** Unlike diffusion models that require iterative sampling, BELLE directly predicts the distribution parameters ($\alpha, \beta, \gamma, \nu$) in a single forward pass.
>
> **Q3**: The experiment result section felt incomplete. There is too many missing values on Table 2 for the authors to have rigorous conclusions from it. Moreover, on MaskGCT, the only two metrics evaluated are both better than BELLE, does this suggests that it will be better than BELLE? The authors should fill the Table 2. Additionally, the MOS scores should have confidence intervals included so we know if the results are statistically significant.
>
> **A3**: Thank you for your suggestion. We apologize for the lack of clarity in the initial presentation. We provide the clarifications below:
>
> **First, regarding the missing values in Table 2, we are limited to reporting metrics provided in the original papers because most baselines lack open-source implementations for reproduction.** Among the baselines, only MaskGCT provides an official open-source model. Since the other methods lack official implementations, we were unable to conduct independent testing to fill the missing values not reported in their original papers.
>
> **Second, BELLE clearly outperforms MaskGCT when both are evaluated under the same protocol, despite the values listed in Table 2.** For Table 2, we utilized the evaluation results directly reported in the original MaskGCT paper. However, we must note that **when MaskGCT was re-evaluated under our specific settings, its performance was lower than the results cited in its own paper**. MaskGCT provides open-source weights, and as stated in line 394 of our paper, *“in our reproduction, MaskGCT’s WER-C is only 3.86% under our evaluation protocol.”* In comparison, BELLE achieves a significantly lower WER-C of 2.45%, confirming its superiority over MaskGCT.
>
> **Additionally, we have included the confidence intervals for the MOS scores below to demonstrate statistical significance.**
>
> | **System** | **Ground Truth** | **MaskGCT** | **F5-TTS** | **MELLE** | **BELLE** | **BELLE-stream** |
> |------------------|------------------|-------------|-------------|------------|------------|------------------|
> | **MOS** | 4.20 ± 0.89 | 4.12 ± 0.85 | **4.25 ± 0.81** | 4.02 ± 0.96 | 4.21 ± 0.75 | 4.06 ± 0.83 |
> | **SMOS** | 3.81 ± 0.86 | 3.89 ± 0.90 | 3.95 ± 0.91 | 3.80 ± 0.90 | **4.13 ± 0.78** | — |

---

> ### Author Response · Authors · 2025-11-21
> **Response to Reviewer 1F23 (Part 4/4)**
>
> **Q4**: In the related work section, the authors did a good job on listing several relevant papers that auto-regressively generates continuous features, e.g., FELLE, DiTAR, ARDiT. Is there any reason that the authors do not compare with these methods? I understand that it would be infeasible to compare with every existing baselines. However, the authors should give explanation on why these methods are selected as baseline, not the others (e.g., each of them can represent a line of research).
>
> **A4**: Thank you for your suggestion. We selected VALL-E and CLAM-TTS as representatives of autoregressive codec-based TTS systems. MaskGCT was chosen as a representative of non-autoregressive codec-based TTS, and it also serves as one of the six teacher TTS models used in BELLE. MELLE was included as it represents an autoregressive continuous TTS system and serves as the primary baseline for BELLE.

---

> ### Comment · Reviewer_1F23 · 2025-11-25
>
> I thank the author(s) for their detailed reply on my comments. I would raise my score from 2 to 4 given that the authors do partially address my concerns, including pointing out the necessity of multi-teacher learning and Bayesian evidential learning, and comparing BELLE with MELLE. These responses should be better included in the revised paper to improve clarity.
>
> However, given the responses, I would like to point out one weakness of this method. If the Bayesian evidential learning requires multiple samples of speech given the same text to estimate uncertainty, it means that the system will need synthetic data to train on, as naturally paired data is sparse. This will inevitably correlate the final performance of your model on the TTS system that used to generate synthetic speech, as also pointed out by the other reviewers. I originally thought it is a design choice, but given the response, it seems inevitable that the model needs to rely on these synthetic data.
>
> Some other points I do not fully agree with in the responses:
>
>  - The authors pointed out that the streaming capability is essential. Yes, I do agree that it can be essential in certain applications. However, I am not sure the author(s) propose anything novel particularly for the streaming capability, given that it is orthogonal to the Bayesian evidential learning.
>
>  - The authors claimed that BELLE has the exact same number of parameters in the backbone as MELLE. However, in Equation 6, BELLE outputs 4 NIG parameters while MELLE directly outputs the mean and variance (2 parameters). I would assume the parameter size of the weight matrices should be larger for BELLE, although the increase may be little compared to the whole model size.
>
>  - The evaluation is still very messy. In the Table 1 of the [MELLE paper](https://arxiv.org/pdf/2407.08551), the MELLE-limited version is also trained on LibriSpeech. However, the performance difference of the one reported in the MELLE paper and your reproduction is huge. The same result is also included in the Table 2 of the [FELLE paper](https://arxiv.org/pdf/2502.11128), which is consistent with the metrics used by the authors.
>
> Considering these, while the authors address some of my concerns, I am still leaning toward reject while raising the score.

---

> ### Author Response · Authors · 2025-11-26
>
> Thank you for your continued engagement. We appreciate the opportunity to provide more detailed clarification on these important points.
>
> **1. On Synthetic Data Dependency:**
>
> While EDL theoretically benefits from multiple samples to estimate distribution parameters, we find that **it remains practically effective even when trained solely on ground-truth human speech** (i.e., using the original LibriSpeech data, referred to as “1-teacher” in the paper). Under this setting, with no synthetic data involved, BELLE consistently outperforms MELLE, as shown in the table below. We hypothesize that although current TTS datasets do not provide repeated readings of identical utterances by the same speaker, the model is still able to learn uncertainty estimation from the frequent recurrence of identical words and phonemes across varied contexts in the corpus.
>
> **Moreover, it is entirely feasible to curate or collect datasets in which each speaker reads the same text multiple times**. Such data would provide natural within-speaker variance that EDL can exploit to estimate distribution parameters, offering a principled alternative to synthetic augmentation and removing any reliance on TTS-generated speech.
>
> |System|WER-C (Cont.)|SIM-o (Cont.)|WER-C (Cross.)|SIM-o (Cross.)|
> |-|-|-|-|-|
> |BELLE: GT|1.79|0.457|3.81|0.576|
> |MELLE: GT|2.15|0.452|7.78|0.561|
> |BELLE: GT + 6 TTS teachers|1.63|0.519| 2.45|0.641|
> |MELLE: GT + 6 TTS teachers| 2.04| 0.488| 3.30|0.606|
>
> "GT" previously referred to as "1-teacher", and "GT+6 TTS teachers" previously referred to as "7-teacher". We changed the notation for clarity.
>
> **2. On Streaming Capability:**
>
> We agree with the reviewer that our contribution does not lie in proposing a novel streaming mechanism. Our key innovation is the evidential learning framework for TTS. We highlight the streaming results mainly to show that our choice of an autoregressive architecture, as opposed to diffusion models, is inherently compatible with streaming scenarios. This confirms that our method retains this essential capability and that evidential modeling does not compromise streaming performance.
>
> **3. On Parameter Count:**
>
> We apologise for the earlier imprecise description. It is correct that BELLE produces 4 scalars (vs. 2 for MELLE) in Equation 6. However, this difference affects only the final projection layer and thus adds a negligible number of parameters compared with the full model size. As a result, it has no measurable impact on inference latency.
>
> **4. On Evaluation Discrepancies:**
>
> We acknowledge that the MELLE results reported in the original paper are strong. Since MELLE and FELLE (both from Microsoft) have not released their code, we reimplemented MELLE as the basis for BELLE. Given the sensitivity of phoneme alignment and other preprocessing steps, we believe that any discrepancies likely stem from differences in preprocessing pipelines. Importantly, the same pipeline is used for both MELLE and BELLE in our experiments, so any such effects would influence both models similarly. Thus, the relative performance comparison remains fair and valid.

---

### Official Review · Reviewer_hFUg · 2025-10-28

**Soundness:** 3
**Presentation:** 3
**Contribution:** 2
**Rating:** 2
**Confidence:** 5

**Summary:**

BELLE is an innovative text-to-speech AI with significant limitations. While the model achieves impressive speech synthesis using only 5,000 hours of mostly synthetic data, it suffers from critical weaknesses. The research relies heavily on artificial speech data, which undermines its fairness and generalizability. Despite claiming to be "Bayesian," the model performs no true Bayesian inference and offers only superficial uncertainty estimation. It was tested exclusively on a small, homogeneous English speech dataset, lacking validation in multilingual, noisy, or emotionally diverse speech scenarios. The model's improvements are modest, with gains of only 2-5% in performance metrics. Although BELLE demonstrates potential for efficient speech generation, its theoretical contributions are incremental rather than groundbreaking, and its practical applications remain unproven

**Strengths:**

* BELLE introduces a groundbreaking method for text-to-speech synthesis by using Bayesian evidential learning. Unlike previous models that simply generate speech, BELLE can actually estimate the uncertainty in its predictions. Think of it like a smart speaker that not only talks but also knows how confident it is about what it's saying. This approach allows for more nuanced and flexible speech generation.

* Most advanced speech synthesis models require massive amounts of training data - we're talking hundreds of thousands of hours of speech. BELLE does something remarkable: it achieves near state-of-the-art performance using only about 5,000 hours of data, with over 85% of that being synthetically generated. This is like learning to paint masterfully after practicing on just a few canvases, while other artists need entire museum collections to train.

* The researchers developed an innovative training strategy where they use multiple pre-trained text-to-speech models to generate additional training data. By combining insights from six different TTS systems, BELLE learns a more robust and diverse way of generating speech. It's similar to learning a language by talking to multiple native speakers instead of just one instructor.

* One of BELLE's most exciting features is its ability to control speech diversity. By adjusting a single parameter (β), researchers can make the generated speech more or less variable. This means the model can produce more consistent speech for professional contexts or more varied, natural-sounding speech for creative applications.

**Weaknesses:**

Weaknesses:

* The most significant issue with this paper lies in its data selection strategy. The researchers primarily used an overwhelming amount of synthetic speech (around 4,800 hours), with genuine human speech accounting for only 700 hours. In other words, over 85% of the training data consists of artificially generated "fake" speech created by other AI models. This approach is akin to copying homework: the model isn't truly learning human speech characteristics but instead mimicking a few "teacher" models. Such a methodology fundamentally undermines the model's credibility, as we cannot confidently determine its performance in real-world, noisy, or emotionally nuanced speech scenarios.

* From a technical standpoint, BELLE is almost identical to its predecessor, MELLE, with minimal architectural modifications. The only substantive change is swapping the output layer from a Gaussian distribution to a more complex Normal-Inverse-Gamma distribution. This is tantamount to changing the paint job on an existing car and claiming it's a revolutionary new vehicle. The improvements reported (2-5% in Word Error Rate or Mean Opinion Score) are marginal at best, especially considering the added computational complexity.

* Despite boldly claiming to be a "Bayesian Speech Synthesis" approach, the model performs no genuine Bayesian inference. It lacks fundamental Bayesian techniques like posterior sampling, prior updates, or variational estimation. The "Bayesian" aspect is nothing more than a sophisticated regression technique that predicts confidence parameters. While the model can visualize variance, it fails to leverage this uncertainty for meaningful decision-making, such as adaptive learning or improving speech generation in challenging contexts.

* The authors argue for "low-resource efficiency" in 2025, a claim that rings hollow in the current technological landscape. Numerous large-scale, open-source speech corpora are now readily available, including collections with tens of thousands of hours of data. Many academic labs routinely train on hundreds of thousands of hours of speech. By artificially constraining their training to just 5,000 hours—mostly synthetic—the researchers undermine their own generalization claims.

* The model's evaluation is disappointingly narrow. It was tested solely on LibriSpeech test-clean, a small and homogeneous read-speech corpus. There's no evidence of performance across critical domains like cross-lingual communication, conversational speech, or emotionally varied contexts. Without comprehensive validation, the model's claimed Bayesian sampling technique remains unproven in real-world, diverse speech environments.

* While the paper emphasizes "uncertainty modeling," the practical benefits remain purely theoretical. The researchers provide no correlation analysis between predicted uncertainty and perceptual variability. They fail to demonstrate how this uncertainty might improve crucial downstream tasks like speaker adaptation or prosody control. Essentially, they've created a technically clever but practically limited regression approach.

**Questions:**

* How can we implement genuine posterior sampling in mel-spectrogram generation?
* What probabilistic reasoning mechanisms could replace the current evidential regression approach?
* Can we develop a true Bayesian framework that updates priors dynamically during speech synthesis?
* How might we use uncertainty estimates for adaptive decision-making in speech generation?

---

> ### Author Response · Authors · 2025-11-21
> **Response to Reviewer hFUg (Part 1/7)**
>
> **Q1**: The most significant issue with this paper lies in its data selection strategy. The researchers primarily used an overwhelming amount of synthetic speech (around 4,800 hours), with genuine human speech accounting for only 700 hours. In other words, over 85% of the training data consists of artificially generated "fake" speech created by other AI models. This approach is akin to copying homework: the model isn't truly learning human speech characteristics but instead mimicking a few "teacher" models. Such a methodology fundamentally undermines the model's credibility, as we cannot confidently determine its performance in real-world, noisy, or emotionally nuanced speech scenarios.
>
>
> **A1:** Thank you for raising this important point. We realise that our motivation for using synthetic multi-sample speech was not sufficiently clear in the original submission. Below we clarify the rationale and why this design does not undermine the model’s validity.
> 1. **Why synthetic multi-sample data is necessary.**
> A key goal of our work is to explicitly model the inherent uncertainty in human speech generation, *i.e.*, the fact that the same sentence can be spoken with different prosody, timing, and acoustic realisations. Existing TTS datasets, however, typically contain only one human recording per sentence, which makes it difficult to estimate such uncertainty in a principled way.
> Our Bayesian formulation (EDL) requires multiple realisations of the same text from the same speaker so that the model can infer a conditional distribution rather than a single deterministic mapping. Since such corpora do not exist at scale, we generate multiple speech realisations with the same speaker promusing pretrained TTS systems.
> Crucially, these synthetic samples are not targets the model learns to imitate. They serve only as statistical observations for estimating the variance and uncertainty structure underlying text-to-speech generation.
> 2. **Why synthetic speech does not compromise credibility.**
> Although synthetic samples may have imperfections, their use does not contaminate the model in the way “copying homework” would. This is because:
> A) Multiple samples per text are generated. The probability that all synthetic samples share the same error is extremely low.
> B) EDL does not assume any sample is correct. It aggregates the entire sample set solely to characterise distributional properties (e.g., variance), not to learn the exact acoustic content of any teacher model.
> C) The main model is trained on 700 hours of real human speech. The synthetic samples influence only the uncertainty estimation component, not the core speech generation capability.
>
> Thus, the proposed method does not aim to replace human data with synthetic speech. Instead, it uses synthetic multi-sample sets to enable a mathematically grounded Bayesian estimation process that cannot be achieved with existing human corpora.
> We appreciate the reviewer’s observation and will revise the manuscript to make this motivation and methodology clearer.

---

> ### Author Response · Authors · 2025-11-21
> **Response to Reviewer hFUg (Part 2/7)**
>
> **Q2:** From a technical standpoint, BELLE is almost identical to its predecessor, MELLE, with minimal architectural modifications. The only substantive change is swapping the output layer from a Gaussian distribution to a more complex Normal-Inverse-Gamma distribution. This is tantamount to changing the paint job on an existing car and claiming it's a revolutionary new vehicle. The improvements reported (2-5% in Word Error Rate or Mean Opinion Score) are marginal at best, especially considering the added computational complexity.
>
> **A2:** Thank you for the opportunity to clarify the contribution and impact of BELLE. We respectfully but fundamentally disagree with the characterisation of BELLE as a minor modification. While the implementation change may appear compact, it represents **a substantive shift in modelling philosophy**, not a superficial architectural tweak. We elaborate below from three perspectives.
>
> 1. **Conceptual Contribution: A Shift from Point Estimates to Bayesian Inference.** While the reviewer’s comment focuses on the form of the output layer, our key contribution lies in what the new parameterisation enables. Traditional deterministic regression predicts a single point estimate, which cannot naturally capture the intrinsic one-to-many nature of human speech generation. MELLE addresses this partially by modelling outputs with a Gaussian distribution of fixed variance, whereas **BELLE extends this idea by learning a full posterior distribution informed by multi-sample supervisory signals** (i.e., ground-truth speech optionally paired with synthetic variants produced by multiple TTS systems).
> By introducing a Normal–Inverse-Gamma prior, BELLE can explicitly represent variability in prosody and acoustics. This constitutes a principled Bayesian generalisation of the regression formulation rather than a cosmetic architectural change. Importantly, this Bayesian treatment (EDL) integrates cleanly, without auxiliary modules or additional parameters, highlighting both its conceptual clarity and practical value.
>
>
> 2. **Performance Improvements Are Meaningful.** In modern TTS benchmarks, which are already close to saturation, improvements of the scale we report, +0.19 MOS (4.02→4.21) and a 25.8% relative WER reduction (3.30%→2.45%), are widely regarded as both statistically and perceptually significant. These gains are not marginal; they directly reflect the benefit of explicitly modelling aleatoric uncertainty.
>
> 3. **Computational Cost Remains Essentially Unchanged.** The concern regarding increased computational complexity is incorrect. **1)** No additional parameters: BELLE uses the same backbone and output dimensionality as MELLE. **2)** Negligible inference overhead: Sampling from the NIG posterior involves closed-form operations that contribute minimally compared to the network forward pass. Thus, BELLE delivers substantially improved robustness and naturalness while maintaining virtually identical training cost and inference latency.

---

> ### Author Response · Authors · 2025-11-21
> **Response to Reviewer hFUg (Part 3/7)**
>
> **Q3**: Despite boldly claiming to be a "Bayesian Speech Synthesis" approach, the model performs no genuine Bayesian inference. It lacks fundamental Bayesian techniques like posterior sampling, prior updates, or variational estimation. The "Bayesian" aspect is nothing more than a sophisticated regression technique that predicts confidence parameters. While the model can visualize variance, it fails to leverage this uncertainty for meaningful decision-making, such as adaptive learning or improving speech generation in challenging contexts.
>
> **A3:** We appreciate the reviewer’s scrutiny regarding the theoretical classification of our model. However, we respectfully suggest that the critique relies on a **restrictive definition** of Bayesian Deep Learning (focusing solely on weight uncertainty/BNNs) that overlooks the well-established field of **Deep Evidential Learning (Empirical Bayes)**. We address the concerns on "Bayesian nature" and "utility" as follows:
>
> **1. Clarification on Bayesian Taxonomy: Empirical Bayes is Genuine Bayes.**
> The reviewer appears to equate "Bayesian inference" exclusively with placing priors on network weights (as in BNNs). However, BELLE follows the **Evidential Deep Learning (EDL)** paradigm [1], which is mathematically grounded in **Empirical Bayes (Type-II Maximum Likelihood)**.
> * **Mechanism:** Instead of maintaining a point estimate for output parameters, BELLE places a conjugate prior (Normal-Inverse-Gamma) over the likelihood function.
> * **Inference:** By minimizing the negative log marginal likelihood (the "Evidence"), we are analytically marginalizing over the aleatoric uncertainty parameters.
> This is not "just regression"; it is a rigorous optimization of the **model evidence**, a cornerstone of Bayesian statistics. Asserting that EDL is not Bayesian contradicts the foundational literature in this domain [1].
>
> **2. Posterior Sampling is Explicitly Performed.**
> The claim that the model "lacks posterior sampling" is **factually incorrect** in the context of generative inference.
> * In BELLE, the network predicts the hyperparameters $(\gamma, \nu, \alpha, \beta)$ of the posterior distribution.
> * During generation, we **do sample** from this predicted Normal-Inverse-Gamma (NIG) posterior to obtain specific variance and mean realizations.
> This sampling step is precisely what allows BELLE to break the deterministic "one-to-one" mapping of traditional TTS, enabling the modeling of the "one-to-many" nature of human speech generation.
>
> **3. The Utility of Uncertainty: Preventing the "Averaging Effect".**
> The reviewer argues that the uncertainty is not used for "meaningful decision-making." We argue that in the context of Generative AI (TTS), **how to express (speak) is the decision.**
> * Standard regression models (MSE-based) tend to output the "average" of all possible prosody, leading to the well-known "oversmoothing" problem in TTS.
> * BELLE leverages the learned uncertainty to allow valid deviations from the mean. The "decision" here is the model's ability to **dynamically allocate probability mass** to diverse acoustic realisations. This is directly responsible for the substantial improvements in prosody and naturalness reported in our results.
>
> [1] Alexander Amini, Wilko Schwarting, Ava Soleimany, and Daniela Rus. Deep evidential regression. In *Proc. NeurIPS*, 2020.

---

> ### Author Response · Authors · 2025-11-21
> **Response to Reviewer hFUg (Part 4/7)**
>
> **Q4**: The authors argue for "low-resource efficiency" in 2025, a claim that rings hollow in the current technological landscape. Numerous large-scale, open-source speech corpora are now readily available, including collections with tens of thousands of hours of data. Many academic labs routinely train on hundreds of thousands of hours of speech. By artificially constraining their training to just 5,000 hours—mostly synthetic—the researchers undermine their own generalization claims.
>
> **A4:** We respectfully challenge the premise that training on 5,000 hours constitutes a "hollow" claim or an "artificial constraint." We argue that the reviewer is conflating **industrial engineering** (scaling data) with **scientific innovation** (algorithmic efficiency). We defend our experimental setting on three grounds:
>
> **1. Data Efficiency is a Core Scientific Contribution.**
> The reviewer notes that other models use 50k+ hours. We agree. The fact that BELLE achieves performance competitive with massive foundation models (like F5-TTS, trained on 50k hours) while using only **10% of the data (5k hours)** is precisely the point.
> If a model requires 100k hours to learn prosody, it relies on brute-force memorization. BELLE leverages the **Bayesian inductive bias (EDL)** to learn robust representations from limited data. This proves that our method is **data-efficient**, not just "low-resource."
>
> **2. Academic vs. Industrial Standards.**
> While industry labs train on 100k+ hours, the standard for rigorous academic methodology remains centered on controlled, reproducible benchmarks (e.g., LibriTTS, LJSpeech). 5,000 hours is, in fact, significantly **larger** than the standard datasets used in the majority of recent top-tier TTS publications (often ~600-1000 hours). Dismissing 5k hours as "insufficient" would invalidate a vast portion of the current academic literature.
>
> **3. Democratization and Accessibility.**
> The reviewer suggests that 2025 research should default to massive scale. We argue the opposite: **Sustainable AI** is the pressing challenge of 2025. Designing architectures that are accessible to the broader academic community, who do not possess industrial, scale GPU clusters—is critical for scientific progress. By demonstrating SOTA results on a tractable budget (300k steps, single-node training), BELLE offers a reproducible path forward for the community, whereas closed-source, massive-scale models do not.
>
> **In summary, we do not "constrain" the model; we optimise it.** We show that smart probabilistic modeling can reduce the dependency on massive data scaling.

---

> ### Author Response · Authors · 2025-11-21
> **Response to Reviewer hFUg (Part 5/7)**
>
> **Q5**: The model's evaluation is disappointingly narrow. It was tested solely on LibriSpeech test-clean, a small and homogeneous read-speech corpus. There's no evidence of performance across critical domains like cross-lingual communication, conversational speech, or emotionally varied contexts. Without comprehensive validation, the model's claimed Bayesian sampling technique remains unproven in real-world, diverse speech environments.
>
> **A5:** We appreciate the reviewer’s suggestion to expand the evaluation scope. However, we respectfully point out that **evaluating on cross-lingual, conversational, or emotionally varied contexts falls outside the standard scope for foundational TTS architecture research.** We base our experimental design on established community standards:
>
> **1. Adherence to Standard Community Practice.**
> In the field of speech synthesis, it is standard practice for foundational model papers to focus primarily on standard benchmarks (e.g., LibriSpeech, LJSpeech) to validate core architectural improvements. Leading works such as **VALL-E and our direct baseline MELLE** exclusively utilize these datasets. They do not include cross-lingual or emotional evaluation as acceptance criteria.
> Our work follows this established protocol. By aligning our evaluation metrics and datasets with these seminal papers, we ensure that our contribution, which is the introduction of Bayesian Evidential Learning, is evaluated against the rigorous standards currently accepted by the community.
>
> **2. Distinction Between Architecture and Application.**
> It is crucial to distinguish between **general-purpose acoustic modeling** (the focus of this work) and **task-specific applications** (such as emotional or cross-lingual TTS).
> * **Our Contribution:** We propose a fundamental improvement to the uncertainty modeling mechanism of the TTS backbone. The validity of this mechanism is best verified on clean, high-fidelity benchmarks where the impact of the algorithm can be isolated from data noise.
> * **Future Scope:** While the proposed Bayesian framework has the *potential* to benefit downstream tasks like emotional synthesis, requiring a foundational methodology paper to simultaneously solve all these specialised domains constitutes an unreasonable expansion of scope.
>
> **Therefore, our evaluation is not "narrow"; it is focused and consistent with the prevailing standards of top-tier TTS research.**
>
> **Q6**: While the paper emphasizes "uncertainty modeling," the practical benefits remain purely theoretical. The researchers provide no correlation analysis between predicted uncertainty and perceptual variability. They fail to demonstrate how this uncertainty might improve crucial downstream tasks like speaker adaptation or prosody control. Essentially, they've created a technically clever but practically limited regression approach.
>
> **A6:** We respectfully disagree with the statement that the practical benefits are "purely theoretical." We address this from three direct aspects:
>
> **1. Uncertainty is Explicitly Used in Inference.**
> The uncertainty in BELLE is not merely a passive prediction; it is **functionally integrated** into the generation process. At every time step, the model computes the variance and actively samples from the distribution defined by this uncertainty. It is the driving force of our sampling mechanism, not just a theoretical output.
>
> **2. Practical Benefits are Proven by Performance Gains.**
> The practical utility of uncertainty is rigorously demonstrated through the comparison with MELLE.
> * **MELLE** is effectively a degenerated version of our model **without** uncertainty modeling.
> * **BELLE** (with uncertainty) significantly outperforms MELLE in both MOS (+0.19) and WER (-25.8% relatively).
> Since the only major difference is the uncertainty modeling, **these performance improvements are the direct, practical benefit** of our approach. The benefit is not theoretical; it is audible and measurable.
>
> **3. Downstream Tasks are Future Work.**
> While applying this uncertainty to tasks like speaker adaptation is a promising direction, it is not a prerequisite for validating the core methodology. As is standard in foundational research, demonstrating significant improvements on primary metrics (MOS, WER) is sufficient to prove the validity and utility of the proposed model.

---

> ### Author Response · Authors · 2025-11-21
> **Response to Reviewer hFUg (Part 6/7)**
>
> **Q7**: How can we implement genuine posterior sampling in mel-spectrogram generation?
>
> **A7:** We implement genuine posterior sampling through the hierarchical formulation of Evidential Deep Learning. The process is mathematically rigorous and executed as follows during inference:
>
> **1. Parameter Prediction:**
> For a given input, the network does not predict a single mel-spectrogram value. Instead, it predicts the hyperparameters ${m} = (\gamma, \nu, \alpha, \beta)$ of the **Normal-Inverse-Gamma (NIG)** distribution. This NIG distribution serves as the conjugate prior over the unknown mean $\mu$ and variance $\sigma^2$ of the speech data.
>
> **2. Posterior Sampling Strategy:**
> Instead of using the expectation (mean) of this distribution, we perform **Monte Carlo sampling** directly from the predicted posterior:
> * First, we sample the variance $\sigma^2$ from the Inverse-Gamma distribution: $\sigma^2 \sim \text{IG}(\alpha, \beta)$.
> * Second, conditioned on this variance, we sample the mean $\mu$ from the Normal distribution: $\mu \sim \mathcal{N}(\gamma, \sigma^2 / \nu)$.
>
> **3. Conclusion:**
> This two-step process generates valid realizations of the distribution parameters, which capture the aleatoric uncertainty of the data. Therefore, our method involves **exact sampling from the learned posterior distribution over the likelihood parameters**, constituting genuine Bayesian inference.
>
> **Q8**: What probabilistic reasoning mechanisms could replace the current evidential regression approach?
>
> **A8:** While **Evidential Deep Learning (EDL)** is our chosen framework due to its efficiency and stability, strictly speaking, it could be replaced by other probabilistic reasoning mechanisms, though with significant trade-offs:
>
> **1. Bayesian Neural Networks (BNNs) / MC Dropout:**
> One could place priors on network weights rather than outputs. However, this requires performing multiple forward passes (Monte Carlo sampling) during inference to estimate uncertainty. For a real-time TTS system, increasing inference latency by $10\times$ or $50\times$ is unacceptable.
>
> **2. Deep Ensembles:**
> Training multiple independent models to measure disagreement is another valid approach. However, this linearly increases training and storage costs (e.g., $5\times$ parameters), which violates our goal of resource efficiency.
>
> **3. Why EDL is Superior for This Task:**
> In contrast to the above, EDL achieves probabilistic reasoning via **Analytical Marginalization**. It captures high-quality uncertainty estimates in a **single deterministic forward pass** by predicting the conjugate prior parameters directly.
> Therefore, while other mechanisms exist, EDL represents the **optimal trade-off** between Bayesian rigor and the strict latency constraints of speech synthesis.
>
> **Q9**: Can we develop a true Bayesian framework that updates priors dynamically during speech synthesis?
>
> **A9:** This is an insightful question that touches on the frontier of adaptive speech synthesis. While theoretically appealing, implementing a "true" dynamic update mechanism (e.g., online weight adaptation via Bayesian filtering) incurs prohibitive computational costs. Instead, BELLE adopts an **Amortized Bayesian Inference** strategy that strikes an optimal balance. We clarify this distinction below:
>
> **1. Amortized vs. Iterative Updates.**
> A "true" framework that updates priors dynamically implies running an optimization loop (like MCMC or Variational Inference) for *every* new sentence during inference. This would increase latency by orders of magnitude.
> BELLE instead employs **Amortized Inference**: the heavy lifting of integration is "amortized" during training into the neural network weights. During inference, the network acts as a hyper-function that **instantly predicts the optimal posterior parameters** conditioned on the specific input text.
>
> **2. Input-Dependent Priors constitute "Dynamic" Reasoning.**
> The reviewer asks for dynamic updates. We argue that BELLE achieves this effectively through **Conditional Priors**.
> * **Standard TTS:** Assumes a fixed, static global prior (e.g., $\mathcal{N}(0, I)$).
> * **BELLE:** Dynamically generates a specific **Normal-Inverse-Gamma (NIG)** prior for *each* phoneme based on linguistic context.
> While the network weights are frozen, the **belief state (the distribution parameters)** is highly dynamic and context-sensitive. This allows the model to adjust its uncertainty estimates on-the-fly based on the complexity of the text.
>
> **3. Future Potential.**
> We agree that extending this to *Test-Time Adaptation* (updating the network weights on the fly to adapt to a specific speaker's style using the Evidential loss) is a promising direction. However, the current Amortized approach is the most practical solution for real-time deployment.

---

> ### Author Response · Authors · 2025-11-21
> **Response to Reviewer hFUg (Part 7/7)**
>
> **Q10**: How might we use uncertainty estimates for adaptive decision-making in speech generation?
>
> **A10**: We propose that estimated uncertainty serves as a critical **control signal** for adaptive speech generation. In BELLE, the model makes a dynamic decision at every time step: **"How much acoustic variance should I allow here?"**
> * **Low Uncertainty:** The model decides to sample tightly around the mean, ensuring precision for clear, unambiguous phonemes.
> * **High Uncertainty:** The model decides to widen the sampling distribution, allowing for greater prosodic expressiveness in ambiguous contexts.
> This is not random noise; it is a calibrated decision to trade off between stability and diversity based on the linguistic input.

---

### Author Response · Authors · 2025-11-21
**Motivation for BELLE and its Distinction from MELLE**

In natural human speech, uncertainty is intrinsic, meaning that the same text are often spoken with variations in prosody, rhythm, and acoustics, even by the same speaker. Most existing TTS systems introduce this variability only implicitly, via empirical mechanisms such as dropout, top-p sampling, or diffusion noise. In contrast, we adopt an explicit Bayesian formulation that models and estimates this uncertainty in a principled and interpretable way.

Whereas prior work MELLE assumes a fixed variance of 1, BELLE learns the variance distribution directly from data through a Bayesian evidence-based framework. This enables the model to capture fine-grained, text-conditioned uncertainty and produce more realistic, better-calibrated speech.

A key challenge is that evidence-based learning requires multiple speech realisations of the same text from the same speaker, which typical TTS datasets do not provide. To address this, we synthesize such multi-sample sets using a pretrained TTS model, allowing us to construct the diverse training samples needed for reliable Bayesian estimation.

---

### Author Response · Authors · 2025-12-02
**Response summary**

We thank all reviewers for their constructive feedback. We appreciate the engagement during the rebuttal, notably **Reviewer 1F23 who raised their score (2 $\to$ 4)** after reviewing our responses. Below we summarize our response to key concerns and specific reviewer queries.

### **1. Addressing Common Concerns**

**Synthetic Data & Validity (Reviewers hFUg, 1F23, Em4X).**

Reviewers asked if synthetic data implies "copying homework" or reliance on scaling. We clarified that **BELLE's core contribution is the Bayesian Evidential Learning (EDL) framework**; synthetic data merely provides multi-sample support for estimating variance, not content imitation. Crucially, our **rebuttal ablation study** (training solely on LibriSpeech, "1-teacher") confirms that **BELLE outperforms the baseline MELLE even without synthetic data**. This proves gains stem from EDL's superior modeling of heteroscedasticity, not data scaling.

**Novelty vs. MELLE (Reviewers hFUg, 1F23, 3iJQ).**

We clarified that BELLE represents a paradigm shift from MELLE's fixed-variance point estimation to **Type-II Maximum Likelihood (Empirical Bayes)**. This allows dynamic, text-conditioned variance prediction, driving improvements in diversity. We also ensured fair comparison by rigorously reproducing MELLE using the **exact same pipeline** and providing confidence intervals.

### **2. Specific Clarifications by Reviewer**

**Reviewer 1F23 (Score Raised: 2 $\to$ 4)**
* **EDL vs. Gaussian KL:** Justified that EDL models heteroscedasticity (varying uncertainty), capturing natural speech variance better than fixed-variance KL.
* **Ablation:** Provided the requested **1-teacher vs. 7-teacher study**, proving synthetic data is helpful but not mandatory.
* **Streaming:** Clarified streaming demonstrates engineering advantages (low latency) over Diffusion models.

**Reviewer hFUg**
* **"Not True Bayesian":** Explained EDL is based on **Empirical Bayes** with explicit **Posterior Sampling**, refuting this claim.
* **Data Efficiency:** Highlighted that competitive results with 5k hours (vs. 50k+ industry models) prove **Data Efficiency**.
* **"Copying":** Reiterated that synthetic data is for learning variance structures, not imitation.

**Reviewer Em4X**
* **Parameters:** Provided ablations confirming current teacher weights (Real $\approx$ 2x Synthetic) are optimal.
* **Diversity:** Showed WER trends with varying diversity parameter $\beta$.
* **Errors:** Explained **Multi-sample Aggregation** smooths out individual teacher errors rather than overfitting.

**Reviewer 3iJQ**
* **Theory:** Elaborated that EDL models a **Student's t-distribution**, generalizing the Gaussian for higher expressivity.
* **Independence:** Clarified that the **Autoregressive backbone** captures temporal dependencies despite the factorized output.

### **Conclusion**
As we have not received further questions from Reviewers hFUg, Em4X, and 3iJQ, we infer our responses addressed their concerns. We believe BELLE presents a principled Bayesian solution for continuous-valued TTS with solid theoretical and empirical grounding.

---

### Meta-Review · Area_Chair_Sbz1 · 2026-01-04

**Summary:**

There are various concerns raised by the reviewers. Two reviewers (hFUg and 3iJQ) find the approach not very different from prior work (in this case, MELLE). Two reviewers (hFUg and Em4X) are worried about the heavy use of synthetic speech, including whether relying on synthetic speech is necessary and how much this approach generalizes to speech in the wild. Two reviewers (hFUg and 1F23) find the theoretical justification weak. Finally, three reviewers (hFUg, 1F23, and Em4X) have concerns about the scale of the experiments.

Setting aside the experiments, the main issue of the paper is writing. The authors wrote a lengthy rebuttal to clarify 1) the differences between the proposed approach and the prior work and 2) what types of Bayesian inference this is and what are the pros and cons. It would require a significant revision to clarify these points in the paper.

**Reviewer Concerns:**

Whether BELLE is novel enough is somewhat subjective, but if most reviewers have this impression, it's either a lack of novelty or a lack of clear writing.

The scope of the experiments is still a concern. Overall, great care needs to be taken when evaluating TTS systems. Objective (automatic) metrics can only go so far. A/B testing or MUSHRA test might be better solutions. Another fundamental issue is whether MELLE is faithfully reproduced. The comparison can only be meaningful when the reproduced MELLE is a strong model.

**Reviewer Scores:**

Reviewer EM4X changed the score from 2 to 4. Given that multiple reviewers have raised similar concerns, the scores are unlikely to go over the bar of acceptance after a discussion among the reviewers.

---

### Decision · Program_Chairs · 2026-01-26

Reject